# An Iteratively Parallel Generation Method with the Pre-Filling Strategy for Document-level Event Extraction

**Guanhua Huang**[1*], **Runxin Xu**[2], **Ying Zeng**[3], **Jiaze Chen**[3†],
**Zhouwang Yang**[1] and **Weinan E**[2]

[1]University of Science and Technology of China [2]Peking University [3]Bytedance
guanhuahuang@mail.ustc.edu.cn, runxinxu@gmail.com, zengying.ss@bytedance.com
teoyde@gmail.com, yangzw@ustc.edu.cn, weinan@math.pku.edu.cn

## Abstract

In document-level event extraction (DEE) tasks, a document typically contains many event records with multiple event roles. Therefore, accurately extracting all event records is a big challenge since the number of event records is not given. Previous works present the entity-based directed acyclic graph (EDAG) generation methods to autoregressively generate event roles, which requires a given generation order. Meanwhile, parallel methods are proposed to generate all event roles simultaneously, but suffer from the *inadequate training* which manifests zero accuracies on some event roles. In this paper, we propose an **I**teratively **P**arallel **G**eneration method with the **P**re-**F**illing strategy (IPGPF). Event roles in an event record are generated in parallel to avoid order selection, and the event records are iteratively generated to utilize historical results. Experiments on two public datasets show our IPGPF improves 11.7 F1 than previous parallel models and up to 5.1 F1 than auto-regressive models under the control variable settings. Moreover, our enhanced IPGPF outperforms other entity-enhanced models and achieves new state-of-the-art performance [1].

## 1 Introduction

Document-level event extraction (DEE) aims to extract multiple event records from the entire document (Ebner et al., 2020; Du et al., 2021b; Li et al., 2021; Xu et al., 2022). Different from sentence-level event extraction (SEE) (Chen et al., 2015; Nguyen et al., 2016; Du and Cardie, 2020b), event arguments of an event record are usually scattered across multiple sentences, while overlapping arguments contained in several event records appear more often. Moreover, real-world event records

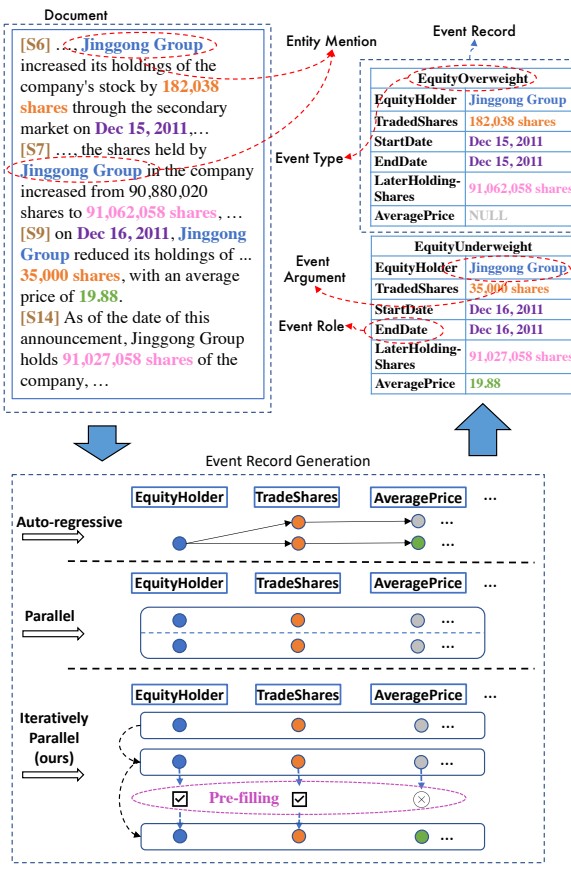

Figure 1: An example of document-level event extraction and different event record generation methods.

always use similar but different words to report the same type of events, leading to low annotations of triggers and the requirement of trigger-free methods. The absence of triggers increases the difficulty of DEE. Therefore, extracting multiple event records without triggers is the main challenge of the trigger-free DEE methods.

Most trigger-free DEE methods (Zheng et al., 2019; Xu et al., 2021a; Liang et al., 2022) build entity-based directed acyclic graph (EDAG) to auto-regressively generate event arguments with their roles under a predefined order. However, it is important to note that determining the optimal order

---

*Work was done when Guanhua was an intern at ByteDance AI Lab.

†Corresponding author.

[1]Our code is available at https://github.com/CarlanLark/IPGPF

from the permutation space of possible orders has an NP-hard complexity of $O(n!)$. Parallel models (Yang et al., 2021) are presented to generate all event records and roles simultaneously to avoid the error broadcasts in a given event role order. Nevertheless, parallel models are particularly prone to the *inadequate training*, leading to zero accuracies on certain event roles.

In this paper, we propose an **I**teratively **P**arallel **G**eneration method with the **P**re-**F**illing strategy (IPGPF). As shown in Figure 1, event roles in an event record are generated in parallel, thereby avoiding the role order selection. Meanwhile, event records are iteratively generated to leverage the historically generated records. The pre-filling strategy is further designed for the *inadequate training* in parallel event role generation. Specially, the pre-filling strategy adaptively pre-fills several event roles by the results in historically generated event records before the parallel event role generation, and then the rest roles are generated with the help of pre-filled roles.

Moreover, we explore the effect of event role orders on auto-regressive EDAG models. Experiments show that a change in event role orders in these models can bring 3.7 F1 decrease over a whole dataset and 10.4 F1 decrease in multiple events scenarios.

We summarize our contributions as follows: (1) We propose IPGPF, an iterative generation method that eliminates the need for selecting the order of event role generation. It achieves this through a two-stage matching that refines generated event records by leveraging historical results. (2) We design the pre-filling strategy, which successfully alleviates the *inadequate training* and zero-accuracy problems in parallel event role generation. (3) Experiments on two large-scale datasets show IPGPF gets 11.7 F1 performance gain than parallel methods and outperforms 1.4 to 5.1 F1 than auto-regressive generation methods in control variable comparison. Additionally, IPGPF is flexible enough to incorporate entity enhancement and outperforms other enhanced models.

## 2 Preliminaries

We first clarify several important concepts: (1) **named entity**: mentions of rigid designators from text belonging to predefined semantic types such as person, location, organization etc (Nadeau and Sekine, 2007); (2) **entity mention**: a text span of

entity in the document which refers to a named entity; (3) **event argument**: an entity playing a specific event role in event extraction; (4) **event role**: a predefined argument type corresponding to event arguments; (5) **event record**: the expression of an event that contains many event arguments with their event roles.

A trigger-free doc-level event extraction task usually includes several pipeline sub-tasks. Following (Zheng et al., 2019), IPGPF finishes the event extraction by handling three sub-tasks: (1) **Named Entity Recognition (NER)**: extracting entity mentions as argument candidates from the document; (2) **Event Detection (ED)**: judging whether there exist predefined event types in the document. (3) **Event Record Generation (ERG)**: generating event records type-by-type. Besides, the model must be able to generate multiple records for one specific event type since there is no trigger.

Due to the space limit, we put the details of **NER** and **ED** sub-tasks that are the same with (Zheng et al., 2019) to Appendix A.3 and A.4.

## 3 Methodology

Given a document $D$ containing $N_s$ sentences $\{s_i\}_{i=1}^{N_s}$, doc-level event extraction aims to generate multiple event records $Z = \{z_i\}_{i=1}^{N_z}$, where $N_z$ is the number of ground truth event records of the document. An event record consists of $n$ event arguments $z_i = (a_i^1, a_i^2, ..., a_i^n)$ and their event roles $(r_i^1, r_i^2, ..., r_i^n)$.

As shown in Figure 2, IPGPF first extracts entity mentions as candidate arguments through named entity recognition (Appendix A.3). The presence of predefined event types is then judged by event detection (Appendix A.4). Finally, event roles in an event record are generated in parallel, while event records are generated iteratively (Section 3.1). To alleviate the *inadequate training* in parallel generation, we propose an effective pre-filling strategy (Section 3.2).

A symbol reminder is listed in Table 7 for a better understanding of our method.

### 3.1 Iteratively Parallel Generation

After the **NER** task and **ED** task, we obtain the candidates arguments representation $H^a \in \mathbb{R}^{(N_a+1)\times d}$ and document sentence representation $H^s \in \mathbb{R}^{N_s \times d}$, where $N_a$ is the number of candidate arguments, $N_s$ is the sentence number, and $d$ is the hidden dimension. Then we generate event records

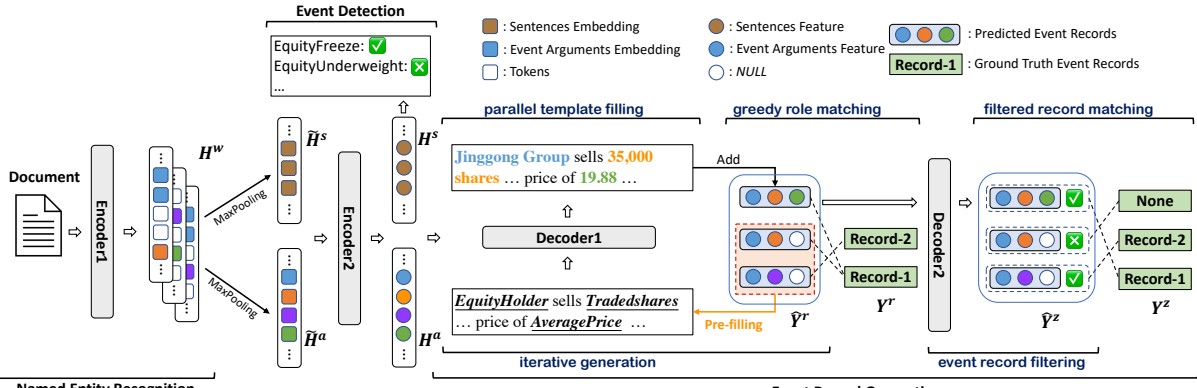

Figure 2: The overall architecture of IPGPF. Given a document, IPGPF first extracts entity mentions as candidate arguments. Then the existence of event types is detected. Finally, event roles are generated from candidate arguments by filling queries in a manual template in parallel, and event records are iteratively generated. A two-stage many-to-one matching loss is further designed for the training of IPGPF.

iteratively and generate event roles in a record in parallel.

### 3.1.1 Parallel Template Filling

Parallel event role generation is an excellent way to avoid unstable performance due to role order selection.

To further help event role generation, we manually build templates for each event type. As shown in Figure 2 and Figure 8, event roles are represented by special tokens in these templates. So we can generate event records by filling the special event role tokens in given templates.

In event record generation of a special event type, we first embed the corresponding template as $Q^t = [q_1^t, q_2^t, ...q_{N_t}^t] \in \mathbb{R}^{N_t \times d}$, where $N_t$ is the number of template tokens. Then a vanilla transformer decoder is employed to encode the template query:

$$H^t = Decoder1(Q^t, H^a) \qquad (1)$$

where $H^t \in \mathbb{R}^{N_t \times d}$, $H^a \in \mathbb{R}^{(N_a+1) \times d}$ is the hidden representation of candidate arguments in Equation (17). Next, we leverage a pointer network (Vinyals et al., 2015):

$$P^r = Softmax(tanh(H^r W^r + H^a W^a) \cdot v) \quad (2)$$

where $H^r \in \mathbb{R}^{N_r \times d}$ is event role representation from $H^t$, $W^r, W^a \in \mathbb{R}^{d \times d}, v \in \mathbb{R}^d$ are trainable parameters, $+$ is the broadcasting plus of two matrices, $P^r \in \mathbb{R}^{N_r \times (N_a+1)}$ is the score of arguments corresponding to roles in the event type.

Finally, we extract arguments $\hat{H}^a \in \mathbb{R}^{N_r \times d}$ corresponding to event roles.

### 3.1.2 Iterative Generation

To better use the generated historical event records, we design an iterative generation method.

As shown in Figure 2, in the iteration, a historical event record is first represented by combining the template queries and extracted arguments: $h^z = MaxPooling([\hat{H}^a, Q^t]) \in \mathbb{R}^d$

At $(i + 1)$-th iteration, all historical records $H^z = [h_1^z, h_2^z, ..., h_i^z] \in \mathbb{R}^{i \times d}$ are concatenated to the template queries in Equation (1) for the template filling task of next iteration:

$$[H^t, \check{H}^z] = Decoder1([Q^t, H^z], H^a) \quad (3)$$

### 3.1.3 Event Record Filtering

As shown in Figure 2, to extract all event records in a document through iterative generation, the iteration number is usually larger than the number of ground truth event records. Therefore, a filter is required to filter several output records as final results.

After $N_i$ iteration, we get event record representation $H^z = [h_1^z, h_2^z, ..., h_{N_i}^z] \in \mathbb{R}^{N_i \times d}$. Then we filter several best records as the final outputs by utilizing a vanilla transformer decoder and a linear layer classifier:

$$\tilde{H}^z = Decoder2(H^z, H^s) \qquad (4)$$

$$P^z = Sigmoid(\tilde{H}^z \cdot W^z) \qquad (5)$$

where $W^z \in \mathbb{R}^d$ are trainable parameters, $P^z \in \mathbb{R}^{N_i}$ are scores of output event scores.

### 3.1.4 Two stage Matching

Different from (Yang et al., 2021) which use a one-stage one-to-one matching to train their parallel

model, we design a two-stage many-to-one matching algorithm to first performs multiple iterative refinements on the generated event records, and subsequently filters the best results.

In the model training, following (Zheng et al., 2019), we generate event records for each event type independently and finally sum the loss of all event types. In this subsection, we will describe the training loss definition for one type in detail.

**Greedy Role Matching** : Given extracted arguments score $\hat{Y}^r = [P_1^r, P_2^r, ..., P_{N_i}^r]$ from Equation (2), and the ground truth arguments $Y^r = [Y_1^r, Y_2^r, ..., Y_{N_{gt}}^r]$, where $Y_i^r = (y_1^r, y_2^r, ..., y_{N_r}^r)$ and $y_j^r$ is the golden label refer to the $j$-th role in $i$-th event record. We define a cost function and pairwise compute the cost between output record roles $\hat{Y}_i^r = P_i^r \in \mathbb{R}^{N_r \times (N_a+1)}$ and ground truth record $Y_j^r \in \mathbb{R}^{N_r}$:

$$C_{role}(\hat{Y}_i^r, Y_j^r) = -\sum_{k=1}^{N_r} Y_{j,k}^r \log P_{i,k}^r \quad (6)$$

We build a greedy many-to-one matching to iteratively refine the event records multiple times, by assigning the most similar ground truth record for each $\hat{Y}_i^r$ as its label:

$$j^* = \arg\min_j C_{role}(\hat{Y}_i^r, Y_j^r) \quad (7)$$

Then we compute a cross-entropy loss for event role generation:

$$\mathcal{L}_{role} = -\sum_{i=1}^{N_i} \sum_{k=1}^{N_r} Y_{\tau(i),k}^r \log P_{i,k}^r \quad (8)$$

where $\tau$ is a surjective function, which computed as mentioned in Appendix A.5

**Filtered Record Matching** : Given the filter scores $\hat{Y}^z = P^z = (p_1^z, p_2^z, ..., p_{N_i}^z) \in \mathbb{R}^{N_i}$ of output event records, and note that the corresponding label from ground truth record $Y^z = (y_1^z, y_2^z, ..., y_{N_{gt}}^z) = \mathbb{I}^{N_{gt}}$. we pairwise compute the cost value between $i$-th output record $\hat{Y}_i^z = p_i^z$ and $j$-th ground truth record $Y_j^z = y_j^z$:

$$C_{event}(\hat{Y}_i^z, Y_j^z) = -y_j^z \log p_i^z = -\log p_i^z \quad (9)$$

To filter several best output records as the final results and abandon the rest, we combine the event filter cost and role cost:

$$C_{all}(\hat{Y}_i, Y_j) = C_{event}(\hat{y}_i^z, y_j^z) + C_{role}(\hat{Y}_i^r, Y_j^r) \quad (10)$$

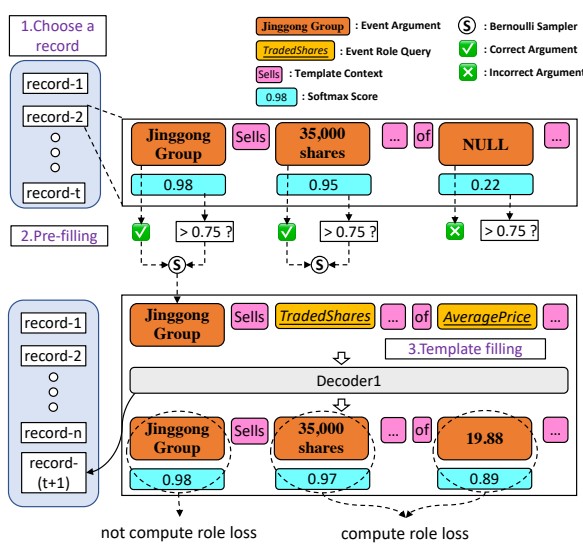

Figure 3: An example of the pre-filling strategy. At $(t+1)$-th iteration step, a historically generated record is first chosen. Then some of its event arguments are pre-filled to corresponding event role queries in a template. Finally, the pre-filled template is fed to a Decoder to fill the rest event role queries.

We find the minimal cost matching $\pi^*$ by the hungarian algorithm (Kuhn, 1955) (Appendix A.5). Then we assign label 1 for records $M$ that matched the ground truth record and label 0 for unmatched records, and get the binary cross-entropy loss:

$$\mathcal{L}_{event} = -(\sum_{i \in M} \log p_i^z + \sum_{i \notin M} \log (1 - p_i^z)) \quad (11)$$

Finally, we combine the role loss and event loss:

$$\mathcal{L}_{ERG} = \gamma_1 \mathcal{L}_{role} + \gamma_2 \mathcal{L}_{event} \quad (12)$$

where $\gamma_1, \gamma_2 \in (0, 1)$ are hyperparameters.

### 3.2 Pre-filling Strategy

Given a document $D$, the goal of DEE model is fitting the joint distribution $P(y_1, y_2, ..., y_n|D)$ of all event roles $\{y_i\}_{i=1}^n$. However, previous parallel methods attempt to learn this joint distribution directly, resulting in a complex and high-dimensional loss landscape, which can susceptiblely lead to *inadequate training* with a failure optimization.

To address this issue, we propose a pre-filling strategy to convert the joint distribution $P(y_1, y_2, ..., y_n|D)$ to a conditional distribution $P(y_i, i \in I_{pf}|D) \times P(y_i, i \notin I_{pf}|D, y_i, i \in I_{pf})$, where $I_{pf}$ represents the index of pre-filled event roles. By doing so, the loss landscape of the conditional distribution is lower-dimensional, bringing a

better parameter optimization. We provide an experimental optimization analysis in Section 4.6.3.

As shown in Figure 3, at $(t+1)$-th iteration, we first check the filtered output event records $P^z = (p_1^z, p_2^z, ..., p_t^z) \in \mathbb{R}^t$ computed by $Decoder2$ in Equation (4), and categorically sample one historical record $\hat{Y}_i$ by probability:

$$p_i^{pf} = \frac{\mathbb{I}(p_i^z > \alpha)(1 - p_i^z)}{\sum_{i=1}^t \mathbb{I}(p_i^z > \alpha)(1 - p_i^z)} \quad (13)$$

where $\alpha$ is the binary classification threshold.

After choosing an output event record, at $(t+1)$-th iteration, we use a Bernoulli sampler to select each correct predicted argument whose score $p^r > \beta$ with probability $\kappa$, where $\beta, \kappa \in (0, 1)$ are hyperparameters. Then we pre-fill these arguments to corresponding event role queries in the template. Finally, the pre-filled template is fed to $Decoder1$ to fill the rest event role queries, and only the loss of not pre-filled queries are computed.

It is notable that our pre-filling strategy is utilized to make a better optimization for model training, inference does not need pre-filling since the model is already trained. Thus, we use ground truth records to select the correct arguments for pre-filling in the model training and do not need them for inference.

### 3.3 Learning

For training our IPGPF model, we make a weighted summing over losses of three sub-tasks:

$$\mathcal{L}_{ALL} = \lambda_1 \mathcal{L}_{NER} + \lambda_2 \mathcal{L}_{ED} + \lambda_3 \mathcal{L}_{ERG} \quad (14)$$

where $\mathcal{L}_{NER}, \mathcal{L}_{ED}, \mathcal{L}_{ERG}$ are losses of **NER**, **ED** and **ERG**, respectively, and $\lambda_1, \lambda_2, \lambda_3$ are hyperparameters of corresponding losses.

### 3.4 Entity Enhancement

The first sub-task of the trigger-free DEE task is **NER**, which enables **ERG** to generate events by selecting arguments from extracted entities. However, in the training data, entity labels are the same as the event argument labels, which brings some entity label conflict issues. For example, the **NER** module can easily recognize "Dec 15, 2011" as a date but confuses about whether it belongs to "*StartDate*" or "*EndDate*", leading to the error propagation.

Thus, we merge several entity labels with the same meaning to a simplified type for the **NER** task. The details of our merge rules can be found in Appendix A.11. Then these simplified entities

are extracted by a pre-trained encoder and used for the next **ERG**. Finally, we get our simple entity-enhanced IPGPF model (IPGPF$^+$) which has better model performance.

## 4 Experiments

### 4.1 Dataset and metrics

The **ChFinAnn** dataset (Zheng et al., 2019) is a large dataset focuses on five event types from the financial text, which has $25,632/3,204/3,204$ for the train/dev/test set. According to statistics, a document in ChFinAnn consists of 20 sentences and has 912 tokens on average, and $29\%$ of $32,040$ documents in ChFinAnn contain multiple event records.

**DuEE-fin** (Han et al., 2022) contains 13 event types, which has $7,015/1,171/59,394$ documents for the train/dev/test set. Especially, the inference results of the test set need to be submitted online for its evaluation. Only $3,513$ documents in the test set are actually evaluated, while the rest $55,881$ documents are given to prevent manual prediction.

Following (Zheng et al., 2019), we use the micro precision, recall, and F1-score over all arguments.

### 4.2 Models for Comparison

**Doc2EDAG** (Zheng et al., 2019) designs an auto-regressive entity-based directed acyclic graph (EDAG) to generate event records. Before the generation, Doc2EDAG uses two transformer encoders to get features of arguments and sentences. **DE-PPN** (Yang et al., 2021) uses the same encoders with Doc2EDAG to obtain the features of arguments and sentences, but generates all event roles in parallel. **SCDEE** (Huang and Jia, 2021) builds an enhanced entity-sentence community graph, then detect event records from the graph and extract entities as arguments corresponding to the event roles. **PTPCG** (Zhu et al., 2022) builds an entity-based pruned complete argument graph with additional entities. It first selects pseudo triggers and then make a beam extraction based on the graph. **GIT** (Xu et al., 2021a) and **RAAT** (Liang et al., 2022) follow the same auto-regressive generation with Doc2EDAG, but add a heterogeneous graph network and entity relation extraction network to enhance the entity representation, respectively.

To investigate the impact of different role orders in auto-regressive models on performance, we conduct a comparison between models with human-selected orders (Doc2EDAG-HS/GIT-HS/RAAT-

HS, using orders in their papers) and models with random-selected orders (Doc2EDAG-RS/GIT-RS/RAAT-RS, reporting the average results of 5 random orders). The detailed orders can be found in Appendix A.13.

Given that **ERG** is the most important and challenging subtask in doc-EE task, we conduct a control variable comparison of Doc2EDAG, DE-PPN and our proposed IPGPF. These models use the same **NER** and **ED** modules but different **ERG** module. We aim to fairly compare our proposed iteratively parallel method with previous auto-regressive and parallel methods.

It is worth noting that although our iterative parallel generation method is mainly proposed for **ERG**, it is flexible to combine with other **NER** and **ED** modules. We also compare our enhanced IPGPF$^+$ with other entity-enhanced models, including SCDEE, PTPCG, GIT, and RAAT, by incorporating a simple entity enhancement mentioned in Section 3.4.

### 4.3 Experiment Settings

We replicated the compared models by their official released code, except for SCDEE which is not open source. Instead, we used the results reported in the SCDEE paper. We provide the details of settings and replications in Appendix A.10.

### 4.4 Main Results

| Models | ChFinAnn | | | DuEE-fin | | |
|---|---|---|---|---|---|---|
| | P | R | F | P | R | F |
| Control Variable Setting | | | | | | |
| Doc2EDAG-HS | 79.0 | 73.8 | 76.3 | - | - | - |
| Doc2EDAG-RS | 81.4 | 67.8 | 73.9 | **67.9** | 46.8 | 55.5 |
| DE-PPN | 76.8 | 72.3 | 74.5 | 56.6 | 38.4 | 45.8 |
| IPGPF (ours) | **82.0** | **73.8** | **77.7** | 64.8 | **51.7** | **57.5** |
| Entity Enhanced Setting | | | | | | |
| SCDEE | **87.2** | 72.0 | 78.9 | - | - | - |
| PTPCG | 83.4 | 74.1 | 78.5 | 65.7 | 54.1 | 59.3 |
| GIT-HS | 82.8 | 76.5 | 79.6 | - | - | - |
| GIT-RS | 83.8 | 72.7 | 77.9 | 65.4 | 53.0 | 58.7 |
| RAAT-HS | 82.9 | **79.3** | 81.1 | **69.2** | 57.4 | 62.8 |
| RAAT-RS | 84.5 | 75.3 | 79.7 | 67.4 | 58.6 | 62.6 |
| IPGPF$^+$ (ours) | 85.7 | 77.3 | **81.3** | 68.2 | **61.8** | **64.8** |

Table 1: F1 scores on the ChFinAnn test set and the DuEE-fin online test set. Doc2EDAG-HS/GIT-HS/RAAT-HS means the results of human selected orders in their paper. Doc2EDAG-RS/GIT-RS/RAAT-RS means the average scores of 5 randomly selected orders for auto-regressive models. Our IPGPF get better performance both on the control variable comparison and the enhanced comparison.

**Overall results** Table 1 presents an overview of the performance of our method on two large-scale public datasets, namely ChFinAnn and DuEE-fin.

In the control variable comparison, our IPGPF significantly outperforms the parallel model DE-PPN by 3.2 F1 on ChFinAnn and 11.7 F1 on DuEE-fin, which is analyzed in detail in Section 4.6. This indicates that our iteratively parallel generation with the pre-filling strategy is better than the pure parallel method. Compared with the auto-regressive model Doc2EDAG, our parallel model IPGPF achieves a 1.4 F1 improvement over the human-selected order and a 3.8 F1 improvement over the average results of randomly selected orders on ChFinAnn. This illustrates that our iterative parallel generation method is more robust than previous auto-regressive methods.

After adding a simple entity enhancement, our IPGPF$^+$ outperforms other entity-enhanced models on both ChFinAnn and DuEE-fin test sets. When compared to the previous state-of-the-art model RAAT, our IPGPF$^+$ outperforms RAAT-HS, which uses human-selected orders, by 0.2 slight better F1 on ChFinAnn and 2.0 F1 on DuEE-fin. Moreover, our IPGPF achieves significant 1.6 improvement on ChFinAnn and 2.2 improvement on DuEE-fin over RAAT-RS, which reports the average performance of randomly selected orders. This reveals that our iteratively parallel generation method is flexible in combining additional features to achieve better performance than stronger SOTA models.

It is worth noting that although RAAT-HS improves by 1.4 on ChFinAnn compared to RAAT-RS, it only improves by 0.2 on DuEE-fin, demonstrating that artificially finding a significantly better order than randomly selected orders is not trivial.

**Single v.s. Multi records** To analyze the model's performance on single and multiple event scenarios, we divided the ChFinAnn test set into two subsets. A document in the single-record set contains only one event record, while a document in the multi-record set contains multiple records.

As shown in the Table 2, IPGPF/IPGPF$^+$ outperforms other models in most single-record scenarios. Although IPGPF/IPGPF$^+$ scored lower than Doc2EDAG-HS/GIT-HS/RAAT-HS on multiple-record scenarios, these auto-regressive models are highly sensitive to the generation order. For instance, Doc2EDAG-RS reported a decrease of 6.8 in multi-record F1 on average compared to Doc2EDAG-HS after randomly changing the role

| Models | EF | | ER | | EU | | EO | | EP | | Overall | |
|---|---|---|---|---|---|---|---|---|---|---|---|---|
| | S. | M. | S. | M. | S. | M. | S. | M. | S. | M. | S. | M. |
| Control Variable Setting | | | | | | | | | | | | |
| Doc2EDAG-HS | 74.8 | **60.7** | 90.5 | 68.5 | 70.1 | 55.9 | 74.4 | 63.2 | 83.1 | **69.1** | 84.2 | **67.4** |
| Doc2EDAG-RS | 75.4 | 58.2 | 88.0 | 65.0 | 74.5 | 58.8 | 75.5 | **64.7** | **85.6** | 60.1 | 84.7 | 60.6 |
| DE-PPN | 70.5 | 58.5 | 90.8 | 66.4 | 70.8 | 53.7 | 68.4 | 53.4 | 82.4 | 66.0 | 83.3 | 64.0 |
| IPGPF (ours) | **82.0** | 57.5 | **93.9** | 68.5 | **76.1** | **60.5** | **79.9** | 58.2 | 85.2 | 67.6 | **87.6** | 66.0 |
| Entity Enhanced Setting | | | | | | | | | | | | |
| SCDEE | - | - | - | - | - | - | - | - | - | - | 88.7 | 65.8 |
| PTPCG | 81.4 | **69.0** | 93.1 | 63.6 | 79.5 | **72.7** | 83.1 | 64.5 | 87.5 | 69.2 | 89.2 | 68.1 |
| GIT-HS | 82.8 | 65.9 | 91.1 | 70.2 | 79.8 | 66.4 | 79.4 | 69.0 | 85.8 | 72.4 | 87.0 | 71.3 |
| GIT-RS | 83.7 | 63.1 | 91.7 | 66.9 | 80.6 | 63.4 | 80.1 | 65.9 | 86.7 | 60.3 | 88.1 | 68.3 |
| RAAT-HS | 75.4 | 66.6 | 93.4 | **73.0** | 80.0 | 68.5 | 78.2 | **74.9** | 86.6 | **74.4** | 87.7 | **73.5** |
| RAAT-RS | 79.6 | 65.0 | 93.4 | 72.5 | 79.6 | 69.3 | 79.9 | 71.4 | 87.9 | 68.9 | 88.6 | 69.1 |
| IPGPF$^+$ (ours) | **85.2** | 64.7 | **96.8** | 67.7 | **82.4** | 68.6 | **83.5** | 66.0 | **88.3** | 71.1 | **91.0** | 70.1 |

Table 2: Comparison of performance on single-record set (S.) and multi-record set (M.) on ChFinAnn. *-HS means the results of autoregressive models with human selected orders in their paper, while *-RS means the average scores of 5 randomly selected orders. Auto-regressive models (*-RS) gets a crash decrease on the multi-record set after changing the generation order of (*-HS). IPGPF significantly outperforms other parallel models, and get more robust performance than auto-regressive models (*-RS). S.: single-record set; M.: multi-record set.

order. Compared with Doc2EDAG-RS / GIT-RS / RAAT-RS which report the average results of the generation orders, our IPGPF/IPGPF$^+$ get significant improvement on both single record scenarios and multiple record scenarios. These comparison results demonstrate the effectiveness and robustness of IPGPF in both single-record and multiple-record scenarios.

## 4.5 Ablation Study

| Model | P | R | F | S. | M. |
|---|---|---|---|---|---|
| IPGPF | 82.0 | 73.8 | 77.7 | 87.6 | 66.0 |
| - matching | -1.2 | -3.2 | -2.3 | -1.1 | -3.3 |
| - pre-filling | -2.3 | -14.3 | -9.6 | -11.4 | -6.9 |
| IPGPF$^+$ | 85.7 | 77.3 | 81.3 | 91.0 | 70.1 |
| -matching | -1.7 | -4.2 | -3.1 | -1.9 | -5.1 |
| -pre-filling | +5.8 | -47.0 | -35.8 | -28.8 | -45.9 |

Table 3: Ablation study of the two stage many-to-one matching and the pre-filling strategy. S.: single-record set; M.: multi-record set.

To reveal the effectiveness of our proposed two stage many-to-one matching, we replace it with the one-to-one matching. Then both IPGPF and IPGPF$^+$ exhibit a significant decrease in performance. This is because the one-to-one matching approach does not allow for iterative refinement of historical generated event records.

In addition, we evaluate the impact of our pre-filling strategy by dropping it. After removing this strategy, both IPGPF and IPGPF$^+$ receive the performance decrease. As discussed in Section 3.2,

the removal of the pre-filling strategy results in a higher-dimensional loss landscape, which can lead to optimization failure and zero accuracies on many event roles. This ultimately leads to lower overall performance. We provide a detailed analysis of this experiment in Section 4.6.

We also conduct the ablation study of the **Effect of Templates** (Appendix A.6) and the **Effect of Iteration Number** (Appendix A.7).

## 4.6 Analysis

In this section, we want to answer the most important questions in the comparison of auto-regressive, parallel and iteratively parallel methods: (1) What is the effect of different orders for auto-regressive models? (Sec 4.6.1) (2) What is the *inadequate training* problem of paralle methods? (Sec 4.6.2) (3) How does our proposed method address the *inadequate training* problem? (Sec 4.6.3)

To better compare the difference between these generation methods, we focus on the control variable comparison between Doc2EDAG, DE-PPN and our IPGPF which use the same **NER** and **ED** modules but different generation methods.

Moreover, we further analyze the speed comparison in Appendix A.8.

### 4.6.1 Effect of Orders

To reveal the effect of different event role orders to EDAG models, we list the comparison between Doc2EDAG models with different event role generation orders in Tabel 4. After the change of event

| Model | P | R | F1 | S. | M. |
|---|---|---|---|---|---|
| Doc2EDAG-HS | 79.0 | 73.8 | **76.3** | 84.2 | **67.4** |
| Doc2EDAG-RS-1 | 82.5 | _65.2_ | 73.0 | 85.5 | _57.0_ |
| Doc2EDAG-RS-2 | 81.5 | 67.2 | 73.6 | 84.3 | 60.7 |
| Doc2EDAG-RS-3 | **83.4** | 66.8 | 74.2 | **85.7** | 59.4 |
| Doc2EDAG-RS-4 | _78.0_ | **74.1** | 76.0 | 84.3 | 66.8 |
| Doc2EDAG-RS-5 | 81.5 | 65.5 | _72.6_ | _83.5_ | 58.9 |
| *-RS Range | 5.4 | 8.9 | 3.7 | 2.2 | 10.4 |
| *-RS Average | 81.4 | 67.8 | 73.9 | 84.7 | 60.6 |

Table 4: Performance of Doc2EDAG models with different event role generation orders. The highest scores are colored in **bold**, while the lowest scores are underlined. The performance of Doc2EDAG fluctuates widely under different orders.

| Event Role | DE-PPN | IPGPF | IPGPF-*w/o pre-filling* |
|---|---|---|---|
| *TotalPledgedRatio* | _0.0_ | **77.7** | $0.0_{(-77.0)}$ |
| *Pledgee* | _0.0_ | 14.8 | $0.0_{(-14.8)}$ |
| *Pledger* | 44.1 | 51.8 | $0.0_{(-51.8)}$ |
| *EventDate* | _0.0_ | 67.8 | $0.0_{(-67.8)}$ |
| *TotalPledgedShares* | 56.5 | **70.1** | $60.4_{(-9.7)}$ |
| *CompanyName* | 70.0 | **78.8** | $71.8_{(-7.0)}$ |
| *Pledge* | 67.7 | **73.4** | $65.7_{(-7.7)}$ |
| *SelfPledgedRatio* | 67.0 | 74.5 | $0.0_{(-74.5)}$ |
| *DisclosureDate* | 66.9 | 74.9 | $0.0_{(-74.9)}$ |
| Overall | 54.4 | **70.0** | $37.5_{(-32.5)}$ |

Table 5: Event role F1 scores in the event type *EquityPledge* (EP) of DuEE-fin. IPGPF leverages the pre-filling strategy to eliminate all zero accuracy problems compared with DE-PPN and IPGPF-*w/o pre-filling*. F1 improvement of IPGPF over IPGPF-*w/o pre-filling* is listed in brackets.

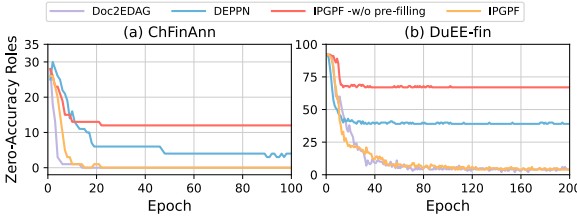

Figure 4: The number of event roles with zero F1 scores on ChFinAnn and DuEE-fin after the model training.

role order, Doc2EDAG-RS-5 reduces 3.7 F1 than Doc2EDAG-HS on the entire ChFinAnn test set, while Doc2EDAG-RS-1 gets a 10.4 F1 crash than Doc2EDAG-HS on multi-record set. On average, Doc2EDAG models with five random event role orders (Doc2EDAG-RS-1 to Doc2EDAG-RS-5) get 1.9 overall F1 lower than Doc2EDAG-HS and 5.7 F1 decrease on multi-record set. These comparisons reveal the EDAG models' heavy reliance on event role orders and significant unstable performance between different event role orders.

### 4.6.2 Inadequate Training

Although parallel generation methods avoid the fluctuation caused by event role orders, these methods encounter a serious *inadequate training* which manifests zero accuracies on many event roles.

Table 5 shows the F1 scores of each event role in the *EquityPledge* (EP) event type on DuEE-fin. For parallel methods, the 0 F1 scores on several event roles of DE-PPN and IPGPF-*w/o pre-filling* clearly reveal the zero accuracy problems of parallel methods due to the *inadequate training* which caused by the failure parameter optimization. Fortunately, the pre-filling strategy successfully alleviates the *inadequate training* and resolves the zero accuracy problems. IPGPF gets good performance on all these zero-accuracy event roles since adding the pre-filling strategy.

As shown in Figure 4, even after training, both DE-PPN and IPGPF-*w/o pre-filling* models have many event roles with zero accuracies. On the ChFinAnn dataset, DE-PPN has 4 event roles with zero accuracy out of 35 total roles. On the DuEE-fin dataset, 35 out of 92 total roles have zero accuracies. Since DuEE-fin has more event roles than ChFinAnn, its joint distribution is higher dimensional, making optimization failure more likely and result-

ing in more zero-accuracy event roles. However, the pre-filling strategy implemented in our IPGPF model solves most of the zero-accuracy problems. The remaining 4 zero-accuracy roles have few data in DuEE-fin, such as the "*Underweight/Holding Ratio*" role with only 20 and 1 non-*NULL* arguments in the training set and dev set, respectively.

### 4.6.3 Optimization Analysis

To illustrate how our pre-filling strategy alleviate the *inadequate training* of parallel models in parameter optimization, we define the gradient norm ratio $R_i^{grad} = \frac{||g_i||^2}{max(||g_j||^2, j=1,2,...,n)}$ to analyze the gradient for model parameters. In detail, $g_i$ is the gradient of the event role $y_i$'s query in these parallel models, such as a $q$ in $Q^t$ in Equation 1.

As shown in the Figure 5 (a)-(c), the event role *"TotalPledgedRatio"* gets zero accuracy for DE-PPN and IPGPF-*w/o pre-filling*. Its gradient norm ratio decreases around zero at the beginning of model training and fails to escape, while the gradient norm ratio corresponding to *"PledgedShares"* remained consistently larger than zero. After adding our pre-filling strategy, *"TotalPledgedRatio"*'s gradient norm ratio avoids the failure optimization and keeps larger than zero significantly, leading to a high F1 score. More details can be

found in Appendix A.14.

We further compare the training loss between our IPGPF and IPGPF-*w/o pre-filling* on DuEE-fin training set. As shown in the Figure 5 (d), after utilizing the pre-filling strategy, the training loss of IPGPF becomes significantly lower than the loss of IPGPF -*w/o pre-filling*, which demonstrates that our pre-filling strategy brings better optimization to parallel models.

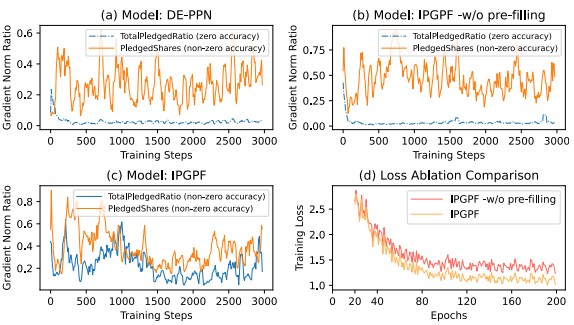

Figure 5: Gradient norm comparison and Training loss ablation comparison.

### 4.7 Case Study

Please refer to Appendix A.9 for the case study.

## 5 Related Work

Most previous works concentrate on extracting events out of a single sentence. Different neural architectures are utilized to extract events, e.g., convolution network (Chen et al., 2015), recurrent network (Nguyen et al., 2016), and Transformer-based network (Yang et al., 2019). Recently, some works also tried to cast the event extraction task as question answering task (Du and Cardie, 2020b; Zhou et al., 2021), or sequence-to-sequence task (Xiangyu et al., 2021). However, these methods can only extract events at the sentence level, which greatly limits their application scenarios.

Therefore, document-level event extraction has been proposed to fully extract events from the entire document. Most document-level EE methods are built upon the event triggers, with which they conduct sequence labelling (Du and Cardie, 2020a; Veyseh et al., 2021) or span-based prediction (Ebner et al., 2020; Zhang et al., 2020; Xu et al., 2022) to identify event arguments in the document. Sequence-to-sequence (Du et al., 2021a,b; Li et al., 2021; Huang et al., 2021; Huang and Peng, 2021; Hsu et al., 2022), and question answering (Wei et al., 2021; Ma et al., 2022) paradigms

are also applied. But the requirement of the annotations of event triggers are quite hard to meet in practice, due to the labeling cost and difficulty.

Consequently, trigger-free methods are required for document-level event extraction. For extracting multiple events without triggers, a trigger-free method is first designed to detect an event in a center sentence and extract the rest of event arguments from surrounding sentences (Yang et al., 2018). Then a widely used entity-based directed acyclic graph (EDAG) generation method is proposed to better deal with multiple events extraction (Zheng et al., 2019). Several variant methods based on EDAG generation are presented by utilizing more meticulous feature engineering, such as heterogeneous graph feature (Xu et al., 2021a; Huang et al., 2021) and entity relation feature (Liang et al., 2022). Additionally, a parallel method is proposed to avoid the error broadcast in EDAG generation (Yang et al., 2021), and an efficient model is designed to lighten the model and accelerates the decoding speed (Zhu et al., 2022).

## 6 Conclusion

Towards the event records extraction in the DEE task, we propose a novel **I**teratively **P**arallel **G**eneration method with the **P**re-**F**illing strategy (IPGPF). IPGPF generates event roles in parallel and iteratively generates event records with the help of the pre-filling strategy. Experiments demonstrate that IPGPF successfully alleviates the *inadequate training* of parallel generation and avoids the event role order selection in auto-regressive generation methods. In the future, we will work on one-stage generation models to avoid the error propagation and time cost of pipeline frameworks.

## Acknowledgements

The work is supported by Anhui Center for Applied Mathematics, NSFC Major Research Plan - Interpretable and General Purpose Next-generation Artificial Intelligence (No. 92270205), and Major Project of Science & Technology of Anhui Province (Nos. 202103a07020011, 202203a05020050).

## Limitations

Based on thousands of words in a document, hidden features obtained from named entity recognition, event detection, and event record generation consumes a large amount of memories. Meanwhile,

the pipeline framework brings more time complexity. Consequently, it costs about 30 to 150 hours on $8 \times 32GB$ GPUs to train Doc2EDAG, DE-PPN, GIT, RAAT, or our IPGPF. Thus, the training cost is the main limitation of our work.

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

# A  Appendix

## A.1  Index

We build an index of each appendix subsections in Table 6 for better understanding our paper.

| Subsection | Title |
|---|---|
| Appendix A.2 | Symbol Reminder |
| Appendix A.3 | Named Entity Recognition |
| Appendix A.4 | Event Detection |
| Appendix A.5 | Two Stage Matching Function |
| Appendix A.6 | Effect of Templates |
| Appendix A.7 | Effect of Iteration Number |
| Appendix A.8 | Speed Comparison |
| Appendix A.9 | Case Study |
| Appendix A.10 | Details of Experiment Settings |
| Appendix A.11 | Details of Entity Enhancement |
| Appendix A.12 | Details of Templates |
| Appendix A.13 | Details of EDAG Orders |
| Appendix A.14 | Details of Optimization Analysis |

Table 6: The index table of appendix subsections.

## A.2 Symbol Reminder

We list the important symbols of methodology into Table 7, for better understanding our approach.

| Symbol | Meaning |
|---|---|
| $d \in \mathbb{N}$ | hidden dimension |
| $N_s \in \mathbb{N}$ | sentence number |
| $N_a \in \mathbb{N}$ | argument number |
| $N_r \in \mathbb{N}$ | role number |
| $N_t \in \mathbb{N}$ | template token number |
| $N_z \in \mathbb{N}$ | event record number |
| $N_{gt} \in \mathbb{N}$ | ground truth record number |
| $N_i \in \mathbb{N}$ | iteration number |
| $N_c \in \mathbb{N}$ | event type number |
| $H^s \in \mathbb{R}^{N_s \times d}$ | sentence representation |
| $H^a \in \mathbb{R}^{N_a \times d}$ | argument representation |
| $H^c \in \mathbb{R}^{N_c \times d}$ | event type representation |
| $H^t \in \mathbb{R}^{N_t \times d}$ | filled template representation |
| $H^z \in \mathbb{R}^{N_z \times d}$ | event record representation |
| $P^c \in \mathbb{R}^{N_c}$ | event type score |
| $P^r \in \mathbb{R}^{N_r \times (N_a+1)}$ | argument score for event roles |
| $P^z \in \mathbb{R}^{N_i}$ | event record filter score |
| $\hat{Y}^r \in \mathbb{R}^{N_i \times N_r \times (N_a+1)}$ | output arguments (score) |
| $Y^r \in \mathbb{N}^{N_{gt} \times N_r}$ | ground truth arguments (index) |
| $\hat{Y}^z \in \mathbb{R}^{N_i}$ | event record filter score |
| $Y^z = \mathbb{I}^{N_{gt}}$ | ground truth record label |

Table 7: The reminder of important symbols.

## A.3 Named Entity Recognition

Given a document $D = \{s_i\}_{i=1}^{N_s}$ that has $N_s$ sentences, IPGPF first embeds words of each sentence as $s = (w_1, w_2, ..., w_{N_w})$ from the document individually, where $N_w$ is the number of words in the sentence. Then these tokens are encoded by a vanilla transformer encoder (Vaswani et al., 2017) to get the hidden representation:

$$H^w = Encoder1([w_1, w_2, ..., w_{N_w}]) \quad (15)$$

Where $H^w \in \mathbb{R}^{N_w \times d}$, $d$ is the hidden dimension. We further add a conditional random field (CRF) layer (Lafferty et al., 2001) after the encoder. Then we employ the BIO (Begin, Inside, Other) tag schema on word representation to extract entity mentions by maximum likelihood labeling:

$$\mathcal{L}_{NER} = -\sum_{s \in D} \sum_{w \in s} \log P(y|w) \quad (16)$$

where $y$ is the label of the word $w$.

## A.4 Event Detection

After sentence-level named entity recognition, we get the entity mentions $E = \{e_i\}_{i=1}^{N_e}$, where $N_e$ is the number of entity mentions in the document.

We obtain the candidate arguments representation $\tilde{H}^a = [\tilde{h}_1^a, \tilde{h}_2^a, ..., \tilde{h}_{N_a}^a, \tilde{h}_{N_a+1}^a] \in \mathbb{R}^{(N_a+1) \times d}$ by max pooling corresponding entity mention representation, $N_a$ is the number of candidate arguments in the document and $\tilde{h}_{N_a+1}^a$ is an additional embedding to represent *NULL* argument. Similarly, we get the sentence representation $\tilde{H}^s = [\tilde{h}_1^s, \tilde{h}_2^s, ..., \tilde{h}_{N_s}^s] \in \mathbb{R}^{N_s \times d}$ by max pooling all word representations in the sentence.

Then we make an interaction between sentence representation and argument representation by employing a vanilla transformer encoder:

$$[H^a, H^s] = Encoder2([\tilde{H}^a, \tilde{H}^s]) \quad (17)$$

For focusing on the comparison of event record generation module, we keep the same feature $H^a \in \mathbb{R}^{(N_a+1) \times d}$ and $H^s \in \mathbb{R}^{N_s \times d}$ with (Zheng et al., 2019; Yang et al., 2021) in Equation (17).

Thus, we can use the argument-aware sentence representation $H^s$ to detect event types by multi-head attention:

$$H^c = Attention(Q^c, [H^a, H^s], [H^a, H^s]) \quad (18)$$
$$P^c = Sigmoid(H^c W^c) \quad (19)$$

where $Q^c \in \mathbb{R}^{N_c \times d}, W^c \in \mathbb{R}^d$ are trainable parameters, $H^c \in \mathbb{R}^{N_c \times d}$ is hidden representation, $P^c \in \mathbb{R}^{N_c}$ is the score of event types, and $N_c$ is the number of predefined event types.

Finally, we train the modules for the event type detection task by optimizing the cross-entropy loss:

$$\mathcal{L}_{ED} = -\sum_{i=1}^{N_c} \mathbb{I}(y_i^c = 1) \log p_i^c$$
$$+ \sum_{i=1}^{N_c} \mathbb{I}(y_i^c = 0) \log (1 - p_i^c)$$

where $y_i^c \in \{0, 1\}$ is the i-th entry of golden label $Y^c \in \mathbb{R}^{N_c}$ and $p_i^c \in (0, 1)$ is the i-th entry of $P^c$.

## A.5 Two Stage Matching Function

**Greedy Role Matching**  After getting the cost function refer to Equation 6

$$C_{role}(\hat{Y}_i^r, Y_j^r) = -\sum_{k=1}^{N_r} Y_{j,k}^r \log P_{i,j}^r$$

We greedily assign the most similar ground truth record for each $\hat{Y}_i^r$ as its label:

$$j^* = \arg\min_j C_{role}(\hat{Y}_i^r, Y_j^r)$$

To inspire our model to generate records that have not been generated in history, we assign labels from ground truth records **without** replacement. After all ground truth records have been assigned, i.e., all records in the document have been discovered, we put all ground truth records back and greedily choose label **with** replacement.

Thus, the greedy many-to-one matching can be described as a surjective function $\{\tau(i) = j, i \in \hat{I}^z, j \in I^z\}$ mapping $i$-th output event record to $j$-th ground truth record, and satisfying $\forall j \in I^z, \exists i \in \hat{I}^z, s.t.\tau(i) = j$.

**Filtered Record Matching** After obtaining the total cost function refer to Equation 10:

$$C_{all}(\hat{Y}_i, Y_j) = C_{event}(\hat{y}_i^z, y_j^z) + C_{role}(\hat{Y}_i^r, Y_j^r)$$

To find the best matching between output event records and ground truth records, we define an injective function $\{\pi(j) = i, j \in I^z, i \in \hat{I}^z\}$ that map index $j$ from ground truth records index set $I^z$ to index $i$ of output records index set $\hat{I}^z$, and satisfying $\{\pi(j_1) \neq \pi(j_2), j_1 \neq j_2, \forall j_1, j_2 \in I^z\}$. Denoting $\Pi(I^z, \hat{I}^z)$ as the set of all injection mappings from $I^z$ to $\hat{I}^z$, we choose the optimal mapping by searching the minimal cost:

$$\pi^* = \underset{\pi \in \Pi(I^z, \hat{I}^z)}{\arg\min} \sum_{j=1}^{N_{gt}} C_{all}(\hat{Y}_{\pi(j)}, Y_j) \quad (20)$$

the minimal cost mapping can be effectively computed by the hungarian algorithm (Kuhn, 1955). Then we can compute the binary cross-entropy loss for event filtering:

$$L_{event} = -(\sum_{i \in A} \log p_i^z + \sum_{i \in B} \log(1 - p_i^z)) \quad (21)$$

where $A = \pi^*(I^z)$ and $B = \hat{I}^z - \pi^*(I^z)$.

## A.6 Effect of Templates

To explore the influence of different template contexts, we change the template contexts for each event type (**IPGPF/IPFPG$^+$-change context**). As shown in Tabel 8, the change of template context just brings a slight performance fluctuation to IPGPF-*change context* compared with IPGPF. Details of our default templates and replacement templates are in Table 8.

To verify the effect of template context, we remove the contexts from templates (**IPGPF/IPFPG$^+$-remove context**) and just

retain the special tokens corresponding to event roles. Results in Tabel 8 show 1.5 F1 decrease on multi-record set compared with IPGPF-*remove context*, which indicates template context is helpful for multiple events scenarios. Details of templates without context can be found in Table 8.

| Model | P | R | F1 | S. | M. |
|---|---|---|---|---|---|
| IPGPF | 82.0 | 73.8 | 77.7 | 87.6 | 66.0 |
| *-change context* | -0.1 | -0.2 | -0.1 | -0.7 | +0.5 |
| *-remove context* | +1.3 | -1.7 | -0.4 | +0.6 | -1.6 |
| IPGPF$^+$ | 85.7 | 77.3 | 81.3 | 91.0 | 70.1 |
| *-change context* | +0.4 | -0.2 | +0.1 | -0.2 | +0.4 |
| *-remove context* | +0.5 | -1.5 | -0.6 | +0.3 | -2.1 |

Table 8: Performance of IPGPF for manual templates. The change of template contexts causes little fluctuation. The removal of the template context reduces the Recall and multi-record set F1-score. S.: single-record set; M.: multi-record set.

## A.7 Effect of Iteration Number

To investigate the effect of the iteration number of IPGPF. We list the model performance of different iteration numbers in Figure 6. Specially, we set the iteration number from $\{1, 3, 5, 7, 9, 11, 13, 15\}$. As shown in Figure 6, after the iteration number larger than 5, the overall F1 score is in the range from 77.5 to 77.7, and the F1 score fluctuation on both single-record set and multi-record set are within 1 point. Even if the iteration number is 15, the model performance does not get a significant decrease. The slight fluctuation during the increase of the iteration numbers suggests the robustness of our IPGPF model.

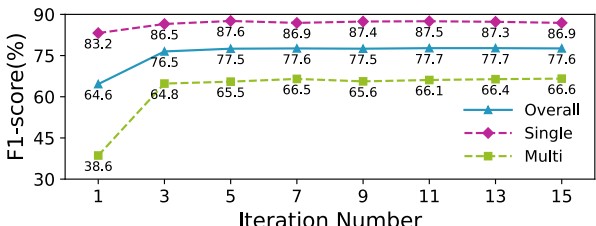

Figure 6: The performance of IPGPF for different iteration numbers. Overall: the entire ChFinAnn test set; Single: single-record set; Multi: multi-record set. The performance on all datasets improves rapidly as the number of iterations increases and stabilizes after the number of iterations is greater than 5.

## A.8 Speed Comparison

We list the comparison of training and inference speed in Table 9. As auto-regressive generation

models, Doc2EDAG, GIT, and RAAT are the slowest in both training and inference speed, whereas the efficient model PTPCG and the parallel model DE-PPN have faster speed, but their performance is relatively lower than other models.

In comparison, our IPGPF$^+$ achieves significant better performance than the current state-of-the-art model RAAT while also having significantly faster training and inference speed. Additionally, we have found that setting the iteration number of IPGPF / IPGPF$^+$ to 5 (default is 10) is a viable option that sacrifices a slight amount of performance for a much faster training speed (16% of RAAT's training time).

| Model | One Epoch Training (mins) | Total Training (days) | Inference Speed (docs/s) | Overall F1 |
|---|---|---|---|---|
| Doc2EDAG | 15.8 | 2.2 | 13.2 | 55.5 |
| DE-PPN | 3.5 | 0.5 | 24.6 | 45.8 |
| IPGPF | 9.0 | 1.3 | 18.7 | 57.5 |
| IPGPF-5 | 5.6 | 0.8 | 25.3 | 57.3 |
| PTPCG | 2.1 | 0.3 | 45.5 | 59.3 |
| GIT | 17.7 | 2.5 | 9.9 | 58.7 |
| RAAT | 44.6 | 6.2 | 6.1 | 62.8 |
| IPGPF$^+$ | 11.3 | 1.6 | 11.6 | 64.8 |
| IPGPF$^+$-5 | 7.2 | 1.0 | 14.9 | 64.5 |

Table 9: The comparison of training and inference speed between different models. We experiment with all models on the DuEE-fin dataset by 8*32G Tesla V100 GPUs. IPGPF-5 / IPGPF$^+$-5 means the iteration number of IPGPF / IPGPF$^+$ is 5 (default is 10).

## A.9 Case Study

To demonstrate the difference between our iteratively parallel method with auto-regressive methods and parallel methods, we conduct a case study on the auto-regressive model Doc2EDAG, parallel model DE-PPN and our iteratively parallel model IPGPF, which use the same **NER** and **ED** modules but different **ERG** modules. Figure 7 shows the prediction results of different generation methods. The document case contains two *EquityPledge* event records, one is an equity pledge release, and another is an equity re-pledge. For auto-regressive models, after changing the generation order of event roles, Doc2EDAG-RS-1 loses a whole event record than Doc2EDAG-HS. For parallel models, DE-PPN failed to learn the event role *TotalPledgedShares* and always generates NULL for *TotalPledgedShares*, which encounters a zero-accuracy due to the *inadequate training*. After removing the pre-filling strategy, although IPGPF *-w/o pre-filling* leverages the iterative generation

to decrease the incorrect arguments than DE-PPN, it still has the *inadequate training* and misses all arguments for *TotalHoldingRatio*. Fortunately, the usage of our pre-filling strategy alleviates the *inadequate training* in parallel methods, and obtains the best results compared to other models. The only incorrect argument *Jun 27, 2016* is not an actual error, the reason is all pledge release records in ground truth just use *ReleasedDate*, but set *StartDate* and *EndDate* to be NULL.

## A.10 Details of Experiment Settings

**Device and Data** To ensure a fair comparison, we run all experiments on the same 8 Tesla-V100 GPUs under the same version of python dependencies. However, we had to use 8 Tesla-A100 GPUs to reproduce the RAAT model on the DuEE-fin dataset, as it uses too many features and causes an out-of-memory problem on V100 GPUs. All compared models in our experiments use the same train/dev/test split for both ChFinAnn [2] and DuEE-fin [3].

**Model Reproduction** We reproduce the compared models Doc2EDAG [4], DE-PPN [5], PTPCG [6], GIT [7], and RAAT [8] by using the code they released, and use the same parameter settings in their paper. We use the results reported in the SCDEE paper since it is not open-sourced. We use the results of DuEE-fin dataset reported in RAAT paper because it just released the human selected order of ChFinAnn dataset but did not release the order for DuEE-fin dataset.

We note that DE-PPN employs extra training tricks and data augmentation, which are not explicitly mentioned in their paper. In our reproduction of DE-PPN, we adopt the extra training tricks and achieve similar performance as described in their official response. However, we have refrained from using any extra training data, as such data

---

[2] https://github.com/dolphin-zs/Doc2EDAG/blob/master/Data.zip
[3] https://aistudio.baidu.com/aistudio/competition/detail/46
[4] https://github.com/dolphin-zs/Doc2EDAG (commit da4f4bc)
[5] https://github.com/HangYang-NLP/DE-PPN (commit b2bc813)
[6] https://github.com/Spico197/DocEE (commit 3fe44b4)
[7] https://github.com/RunxinXu/GIT (commit 3f91743)
[8] https://github.com/TencentYoutuResearch/EventExtraction-RAAT (commit 7a9a726)

are neither mentioned in the DE-PPN paper nor open-sourced. This decision was taken to maintain consistency with the experimental setup reported in the original paper and to ensure a fair comparison with other models that were not using extra training data.

| Hyperparameters | Value |
|---|---|
| Batch Size | 64 |
| Gradient Accumulation Steps | 8 |
| Training Epochs | 100 / 200 |
| Pre-filling Epochs | 30 |
| Learning Rate | 3e-5 / 5e-5 |
| Hidden Dimension | 768 |
| Dropout Rate | 0.1 |
| $\alpha$ | 0.5 |
| $\beta$ | 0.75 |
| $\gamma_1$ | 0.5 |
| $\gamma_2$ | 1.0 |
| $\lambda_1$ | 0.1 |
| $\lambda_2$ | 0.4 |
| $\lambda_3$ | 0.5 |
| Layers of Encoder1 | 4 |
| Layers of Encoder2 | 4 |
| Layers of Decoder1 | 4 |
| Layers of Decoder2 | 4 |

Table 10: Hyperparameters of our IPGPF model

**Hyperparameters** To mitigate the error-propagation of entity mentions recognized by the model, we leverage the same scheduled sampling (Bengio et al., 2015) as (Zheng et al., 2019) did. In the training process, we use golden entity mentions from epoch 1 to epoch 10. From epoch 10 to epoch 20, we gradually switch the mentions from golden label to model output results, by a linear decrease of proportion from 100% to 0%. In our enhanced model IPGPF$^+$, we use the ChildTuning (Xu et al., 2021b) to adapt the model to downstream tasks.

For the pre-filling strategy, we linearly decrease the pre-filling probability $\kappa$ from 120% to 0% in the first 30 epochs of model training, $\kappa$ will be set to 100% if $\kappa > 100\%$ at the current epoch.

We implement IPGPF under Pytorch (Paszke et al., 2017) based on codes released by (Zheng et al., 2019) and (Yang et al., 2021).

For hyperparameters, we do not tune all the hyperparameters in our experiment. For model comparison, we use the same random seed following (Zheng et al., 2019) for compared models.

For the training epochs, we follow (Zheng et al., 2019; Yang et al., 2021; Xu et al., 2021a) to train 100 epochs for all compared models on the ChFinAnn dataset. We train 200 epochs for all compared models on the DuEE-fin dataset since DuEE-fin has more event and argument types but less training data than the ChFinAnn dataset.

## A.11 Details of Entity Enhancement

Table 11 lists our merge rule of entity labels on the ChFinAnn dataset. To further reduce the missing entity labels, we add additional entities matched through regular expressions (Zhu et al., 2022). Regarding the pre-trained encoder, we utilize LERT (Cui et al., 2022) for ChFinAnn and ERNIE (Sun et al., 2019) for DuEE-fin.

To make a fair comparison of different event generation types, IPGPF uses the same entity annotations and entity labels with Doc2EDAG, DE-PPN. We only merge entities or add additional entities for IPGPF$^+$ when compared with the entity-enhanced models such as SCDEE, PTPCG, GIT and RAAT.

| Raw Entity Type | Simplified Type |
|---|---|
| EquityHolder CompanyName Pledger Pledgee | Person & Organization |
| RepurchaseAmount HighestTradingPrice LowestTradingPrice AveragePrice | Amount |
| StartDate EndDate ReleasedDate ClosingDate UnfrozeDate | Date |
| TotalHoldingShares TotalPledgedShares PledgedShares FrozeShares RepurchasedShares TradedShares LaterHoldingShares | Share |

Table 11: The entity label merge rule for the ChFinAnn dataset. Raw entity labels come from the event argument labels.

## A.12 Details of Templates

We listed the templates for the ChFinAnn dataset in Figure 8. As shown in Figure 8, . In the comparison of different templates corresponding to Table 8, we use the default context to train our IPGPF model, the changed context to train the IPGPF-*change context* model, and the none context to train the IPGPF-*remove context* model. All templates for the model training on ChFinAnn are in Chinese, and here we translate these templates to English for illustration.

## A.13 Details of EDAG Orders

As shown in Figure 12, we list the relation between the event roles in the ChFinAnn dataset and their

| Sentence ID | Text |
|---|---|
| [S5] | On Sep 25, 2018, ZhongXin Inc released the 25,000,000 shares (8.33% of the company's total share capital) that it pledged to ICBC Taizhou Branch on Jun 27, 2016. |
| [S6] | ZhongXin Inc pledged 25,000,000 shares of the company's restricted shares held by it (accounting for 8.33% of the company's total share capital and 16.18% of the company's shares) to ICBC Taizhou Branch for a period From Sep 25, 2018 to Sep 17, 2021, the relevant pledge registration procedures have been completed. |
| [S7] | As of the date of this announcement, ZhongXin Inc holds a total of 154,508,497 shares of the company, accounting for 51.48% of the company's total share capital. |
| [S8] | The total number of pledged shares of ZhongXin Inc is 142,250,000 shares, accounting for 92.07% of the company's shares and 47.39% of the company's total share capital. |

| | | | | Event records of *EquityPledge* (EP): **Ground Truth** | | | | | |
|---|---|---|---|---|---|---|---|---|---|
| Record ID | Pledger | PledgedShares | Pledgee | TotalHoldingShares | TotalHoldingRatio | TotalPledgedShares | StartDate | EndDate | ReleasedDate |
| [R1] | ZhongXin Inc | 25,000,000 shares | ICBC Taizhou Branch | 154,508,497 shares | 51.48% | 142,250,000 shares | NULL | NULL | Sep 25, 2018 |
| [R2] | ZhongXin Inc | 25,000,000 shares | ICBC Taizhou Branch | 154,508,497 shares | 51.48% | 142,250,000 shares | Sep 25, 2018 | Sep 17, 2021 | NULL |

| | | | | Event records of *EquityPledge* (EP): **Doc2EDAG-HS** | | | | | |
|---|---|---|---|---|---|---|---|---|---|
| Record ID | Pledger | PledgedShares | Pledgee | TotalHoldingShares | TotalHoldingRatio | TotalPledgedShares | StartDate | EndDate | ReleasedDate |
| [R1] | ZhongXin Inc | 25,000,000 shares | ICBC Taizhou Branch | 154,508,497 shares | 51.48% | NULL | Jun 27, 2016 | NULL | Sep 25, 2018 |
| [R2] | ZhongXin Inc | 25,000,000 shares | ICBC Taizhou Branch | 154,508,497 shares | 51.48% | NULL | Sep 25, 2018 | Sep 17, 2021 | NULL |

| | | | | Event records of *EquityPledge* (EP): **Doc2EDAG-RS-1** | | | | | |
|---|---|---|---|---|---|---|---|---|---|
| Record ID | Pledger | PledgedShares | Pledgee | TotalHoldingShares | TotalHoldingRatio | TotalPledgedShares | StartDate | EndDate | ReleasedDate |
| [R1] | ZhongXin Inc | 25,000,000 shares | ICBC Taizhou Branch | 154,508,497 shares | 51.48% | 142,250,000 shares | Sep 25, 2018 | NULL | NULL |
| - | - | - | - | - | - | - | - | - | - |

| | | | | Event records of *EquityPledge* (EP): **DE-PPN** | | | | | |
|---|---|---|---|---|---|---|---|---|---|
| Record ID | Pledger | PledgedShares | Pledgee | TotalHoldingShares | TotalHoldingRatio | TotalPledgedShares | StartDate | EndDate | ReleasedDate |
| [R1] | ZhongXin Inc | 25,000,000 shares | ICBC Taizhou Branch | 154,508,497 shares | 51.48% | NULL | Sep 25, 2018 | NULL | NULL |
| [R2] | ZhongXin Inc | 25,000,000 shares | ICBC Taizhou Branch | 154,508,497 shares | 51.48% | NULL | Jun 27, 2016 | Sep 17, 2021 | NULL |

| | | | | Event records of *EquityPledge* (EP): **IPGPF-*w/o pre-filling*** | | | | | |
|---|---|---|---|---|---|---|---|---|---|
| Record ID | Pledger | PledgedShares | Pledgee | TotalHoldingShares | TotalHoldingRatio | TotalPledgedShares | StartDate | EndDate | ReleasedDate |
| [R1] | ZhongXin Inc | 25,000,000 shares | ICBC Taizhou Branch | 154,508,497 shares | NULL | 142,250,000 shares | Sep 25, 2018 | NULL | NULL |
| [R2] | ZhongXin Inc | 25,000,000 shares | ICBC Taizhou Branch | 154,508,497 shares | NULL | 142,250,000 shares | Jun 27, 2016 | Sep 17, 2021 | NULL |

| | | | | Event records of *EquityPledge* (EP): **IPGPF** | | | | | |
|---|---|---|---|---|---|---|---|---|---|
| Record ID | Pledger | PledgedShares | Pledgee | TotalHoldingShares | TotalHoldingRatio | TotalPledgedShares | StartDate | EndDate | ReleasedDate |
| [R1] | ZhongXin Inc | 25,000,000 shares | ICBC Taizhou Branch | 154,508,497 shares | 51.48% | 142,250,000 shares | Jun 27, 2016 | NULL | Sep 25, 2018 |
| [R2] | ZhongXin Inc | 25,000,000 shares | ICBC Taizhou Branch | 154,508,497 shares | 51.48% | 142,250,000 shares | Sep 25, 2018 | Sep 17, 2021 | NULL |

Figure 7: A Case for model comparison. We show the text of related sentences. Ground truth entity mentions and arguments are colored in blue, while incorrectly predicted arguments are colored in red. Event roles that get zero accuracies on the whole test set are highlighted .

ids. The event role generation orders of the auto-regressive EDAG models in our experiments are listed in Table 13 to Table 18 by their ids.

## A.14 Details of Optimization Analysis

Figure 9, 10, and 11 show the gradient norm ratio of all event role queries of the parallel models DE-PPN, IPGPF-*without pre-filling*, and our IPGPF during the first 3,000 training steps on the DuEE-fin training set which has 13 event types. In detail, each query is a 768-dimensional query tensor in the model parameters corresponding to an event role. Each broken line in the figure corresponds to one event role, the repeated line refers to the overlapped event roles cross different event types. To better show the comparison, we choose a value for the broken line every 100 steps in the first 3,000 training steps, and draw the final line with a moving average of 5.

The gradient norm ratio of query parameters corresponding to zero-accuracy event roles decreases around zero rapidly and fails to escape, while the gradient norm ratio corresponding to non-zero-accuracy event roles is still larger than zero significantly.

After using the pre-filling strategy, our IPGPF solves the zero-accuracy problem for almost all event roles. The reason for the rest 4 event roles with zero accuracies is the few role arguments in the DuEE-fin dataset. For instance, the event role *"Underweight/Holding Ratio"* just has 20 and 1 not *NULL* arguments on the DuEE-fin training set and dev set, respectively. Moreover, these 4 event roles get zero accuracies for all other compared models such as Doc2EDAG.

| Remove Context (IPGPF-remove context) | |
|---|---|
| **Event Type** | **Template** |
| *EquityFreeze* | *UnfrozeDate*  *LegalInstitution*  *FrozeShares*  *EquityHolder*  *StartDate*  *EndDate*  *TotalHoldingShares*  *TotalHoldingRatio*. |
| *EquityRepurchase* | *CompanyName*  *RepurchasedShares*  *HighestTradingPrice*  *LowestTradingPrice*  *ClosingDate*  *RepurchaseAmount*. |
| *EquityUnderweight* | *EquityHolder*  *TradedShares*  *AveragePrice*  *StartDate*  *EndDate*  *LaterHoldingShares*. |
| *EquityOverweight* | *EquityHolder*  *TradedShares*  *AveragePrice*  *StartDate*  *EndDate*  *LaterHoldingShares*. |
| *EquityPledge* | *ReleasedDate*  *Pledger*  *PledgedShares*  *Pledgee*  *StartDate*  *EndDate*  *TotalHoldingShares*  *TotalHoldingRatio*  *TotalPledgedShares*. |

| Default Context (IPGPF) | |
|---|---|
| **Event Type** | **Template** |
| *EquityFreeze* | On *UnfrozeDate*, *LegalInstitution* freezes or unfreezes the *FrozeShares* held by *EquityHolder*. It starts from *StartDate* and ends at *EndDate*. At present, he/she/it still holds *TotalHoldingShares* shares of the company, accounting for *TotalHoldingRatio* of the company's total share capital. |
| *EquityRepurchase* | *CompanyName* repurchases *RepurchasedShares* at the highest price of *HighestTradingPrice* per share and the lowest price of *LowestTradingPrice* per share, the repurchase time is *ClosingDate*, and the total payment amount was about *RepurchaseAmount*. |
| *EquityUnderweight* | *EquityHolder* sells *TradedShares* of the company at the average price of *AveragePrice* per share. The sale starts from *StartDate* and ends at *EndDate*. After the sale, he/she/it still holds *LaterHoldingShares* shares of the company. |
| *EquityOverweight* | *EquityHolder* buys *TradedShares* of the company at the average price of *AveragePrice* per share. The purchase starts from *StartDate* and ends at *EndDate*. After the purchase, he/she/it still holds *LaterHoldingShares* shares of the company. |
| *EquityPledge* | On *ReleasedDate*, *Pledger* pledges or releases *PledgedShares* to *Pledgee*. It starts from *StartDate* and ends at *EndDate*. He/She/It still holds *TotalHoldingShares* of the company, accounting for *TotalHoldingRatio* of the company's total share capital, and accumulatively pledges *TotalPledgedShares* shares. |

| Changed Context (IPGPF-change context) | |
|---|---|
| **Event Type** | **Template** |
| *EquityFreeze* | From *StartDate* to *EndDate*, *FrozeShares* of *EquityHolder* were frozen or unfrozen by *LegalInstitution*. The unfrozen date is *UnfrozeDate*. At present, he/she/it still holds *TotalHoldingRatio* of the company's total share capital which are *TotalHoldingShares* in total |
| *EquityRepurchase* | On *ClosingDate*, *RepurchasedShares* are repurchased by *CompanyName*. The highest price is *HighestTradingPrice* and the lowest price is *LowestTradingPrice*. *RepurchaseAmount* paid for this repurchasement in total. |
| *EquityUnderweight* | From *StartDate* to *EndDate*, *TradedShares* are sold by *EquityHolder* at the average price of *AveragePrice* per share. After the sale, he/she/it still holds *LaterHoldingShares* shares of the company. |
| *EquityOverweight* | From *StartDate* to *EndDate*, *TradedShares* are bought by *EquityHolder* at the average price of *AveragePrice* per share. After the purchase, he/she/it still holds *LaterHoldingShares* shares of the company. |
| *EquityPledge* | From *StartDate* to *EndDate*, *PledgedShares* are pledged or released to *Pledgee* by *Pledger*. The release date is *ReleasedDate*. The cumulative shares for pledgement are *TotalPledgedShares*. *TotalHoldingRatio* of the company's total share capital were still held by he/she/it, which together account for *TotalHoldingShares*. |

Figure 8: Details for templates. The expressions and event role orders differ between default and replacement templates. We further remove all contexts in templates to build the none-template.

| Event Role | ID |
|---|---|
| EquityHolder | 1 |
| FrozeShares | 2 |
| LegalInstitution | 3 |
| TotalHoldingShares | 4 |
| TotalHoldingRatio | 5 |
| StartDate | 6 |
| EndDate | 7 |
| UnfrozeDate | 8 |
| CompanyName | 9 |
| HighestTradingPrice | 10 |
| LowestTradingPrice | 11 |
| RepurchasedShares | 12 |
| ClosingDate | 13 |
| RepurchaseAmount | 14 |
| TradedShares | 15 |
| LaterHoldingShares | 16 |
| AveragePrice | 17 |
| Pledger | 18 |
| PledgedShares | 19 |
| Pledgee | 20 |
| TotalPledgedShares | 21 |
| ReleasedDate | 22 |

Table 12: The relation between event roles in ChFinAnn and their ids.

| Doc2EDAG-HS / GIT-HS / RAAT-HS | |
|---|---|
| **Event Type** | **Event Role Order** |
| EquityFreeze | $1 \to 2 \to 3 \to 4 \to 5 \to 6 \to 7 \to 8$ |
| EquityRepurchase | $9 \to 10 \to 11 \to 12 \to 13 \to 14$ |
| EquityUnderweight | $1 \to 15 \to 6 \to 7 \to 16 \to 17$ |
| EquityOverweight | $1 \to 15 \to 6 \to 7 \to 16 \to 17$ |
| EquityPledge | $18 \to 19 \to 20 \to 4 \to 5 \to 21 \to 6 \to 7 \to 22$ |

Table 13: The event role generation orders for Doc2EDAG-O and GIT-O. Event roles are represented by their ids.

| Doc2EDAG-RS-1 / GIT-RS-1 / RAAT-RS-1 | |
|---|---|
| **Event Type** | **Event Role Order** |
| EquityFreeze | $6 \to 5 \to 2 \to 8 \to 4 \to 3 \to 7 \to 1$ |
| EquityRepurchase | $11 \to 9 \to 14 \to 13 \to 10 \to 12$ |
| EquityUnderweight | $16 \to 6 \to 17 \to 1 \to 15 \to 7$ |
| EquityOverweight | $15 \to 17 \to 1 \to 16 \to 7 \to 6$ |
| EquityPledge | $21 \to 22 \to 20 \to 5 \to 7 \to 4 \to 6 \to 18 \to 19$ |

Table 14: The event role generation orders for Doc2EDAG-1 and GIT-1. Event roles are represented by their ids.

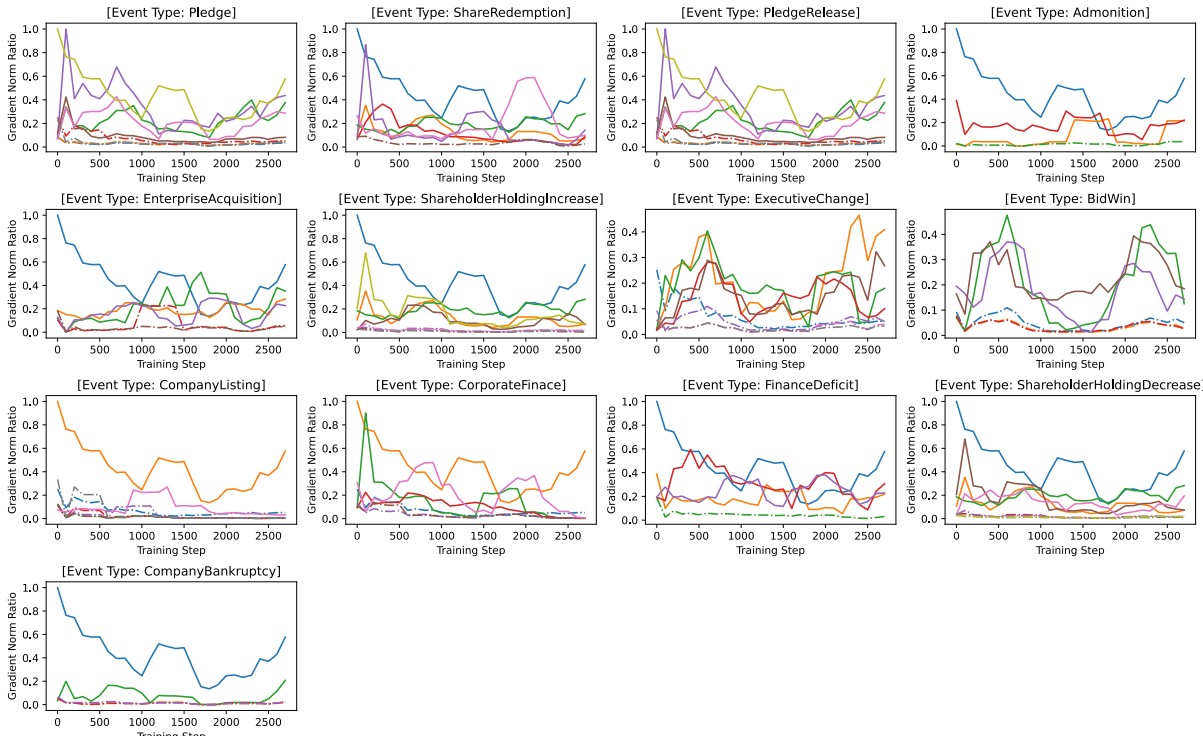

Figure 9: Gradient Norm Ratio Analysis of DE-PPN Model on DuEE-fin Training Set. The plot depicts the gradient norm ratio of the queries associated with each event role in the DuEE-fin dataset during the initial 3,000 training steps. The solid line represents event roles with non-zero accuracy on the dev/test set, while the dashed line denotes event roles with zero accuracy.

| Doc2EDAG-RS-2 / GIT-RS-2 / RAAT-RS-2 | |
|---|---|
| **Event Type** | **Event Role Order** |
| EquityFreeze | $6 \rightarrow 5 \rightarrow 4 \rightarrow 7 \rightarrow 2 \rightarrow 8 \rightarrow 3 \rightarrow 1$ |
| EquityRepurchase | $14 \rightarrow 11 \rightarrow 13 \rightarrow 10 \rightarrow 12 \rightarrow 9$ |
| EquityUnderweight | $7 \rightarrow 16 \rightarrow 17 \rightarrow 6 \rightarrow 15 \rightarrow 1$ |
| EquityOverweight | $17 \rightarrow 15 \rightarrow 6 \rightarrow 16 \rightarrow 1 \rightarrow 7$ |
| EquityPledge | $7 \rightarrow 4 \rightarrow 22 \rightarrow 21 \rightarrow 20 \rightarrow 6 \rightarrow 5 \rightarrow 19 \rightarrow 18$ |

Table 15: The event role generation orders for Doc2EDAG-2 and GIT-2. Event roles are represented by their ids.

| Doc2EDAG-RS-4 / GIT-RS-4 / RAAT-RS-4 | |
|---|---|
| **Event Type** | **Event Role Order** |
| EquityFreeze | $1 \rightarrow 4 \rightarrow 3 \rightarrow 2 \rightarrow 8 \rightarrow 5 \rightarrow 6 \rightarrow 7$ |
| EquityRepurchase | $10 \rightarrow 9 \rightarrow 14 \rightarrow 12 \rightarrow 11 \rightarrow 13$ |
| EquityUnderweight | $15 \rightarrow 1 \rightarrow 6 \rightarrow 16 \rightarrow 17 \rightarrow 7$ |
| EquityOverweight | $1 \rightarrow 15 \rightarrow 17 \rightarrow 7 \rightarrow 16 \rightarrow 6$ |
| EquityPledge | $19 \rightarrow 5 \rightarrow 18 \rightarrow 4 \rightarrow 7 \rightarrow 20 \rightarrow 6 \rightarrow 22 \rightarrow 21$ |

Table 17: The event role generation orders for Doc2EDAG-4 and GIT-4. Event roles are represented by their ids.

| Doc2EDAG-RS-3 / GIT-RS-3 / RAAT-RS-3 | |
|---|---|
| **Event Type** | **Event Role Order** |
| EquityFreeze | $7 \rightarrow 8 \rightarrow 5 \rightarrow 6 \rightarrow 3 \rightarrow 4 \rightarrow 2 \rightarrow 1$ |
| EquityRepurchase | $13 \rightarrow 12 \rightarrow 14 \rightarrow 11 \rightarrow 9 \rightarrow 10$ |
| EquityUnderweight | $17 \rightarrow 6 \rightarrow 7 \rightarrow 1 \rightarrow 16 \rightarrow 15$ |
| EquityOverweight | $7 \rightarrow 16 \rightarrow 17 \rightarrow 15 \rightarrow 6 \rightarrow 1$ |
| EquityPledge | $21 \rightarrow 22 \rightarrow 6 \rightarrow 7 \rightarrow 20 \rightarrow 5 \rightarrow 18 \rightarrow 19 \rightarrow 4$ |

Table 16: The event role generation orders for Doc2EDAG-3 and GIT-3. Event roles are represented by their ids.

| Doc2EDAG-RS-5 / GIT-RS-5 / RAAT-RS-5 | |
|---|---|
| **Event Type** | **Event Role Order** |
| EquityFreeze | $8 \rightarrow 3 \rightarrow 1 \rightarrow 5 \rightarrow 2 \rightarrow 4 \rightarrow 7 \rightarrow 6$ |
| EquityRepurchase | $14 \rightarrow 12 \rightarrow 9 \rightarrow 13 \rightarrow 10 \rightarrow 11$ |
| EquityUnderweight | $17 \rightarrow 15 \rightarrow 1 \rightarrow 16 \rightarrow 7 \rightarrow 6$ |
| EquityOverweight | $7 \rightarrow 6 \rightarrow 1 \rightarrow 17 \rightarrow 16 \rightarrow 15$ |
| EquityPledge | $22 \rightarrow 20 \rightarrow 21 \rightarrow 7 \rightarrow 18 \rightarrow 19 \rightarrow 5 \rightarrow 4 \rightarrow 6$ |

Table 18: The event role generation orders for Doc2EDAG-5 and GIT-5. Event roles are represented by their ids.

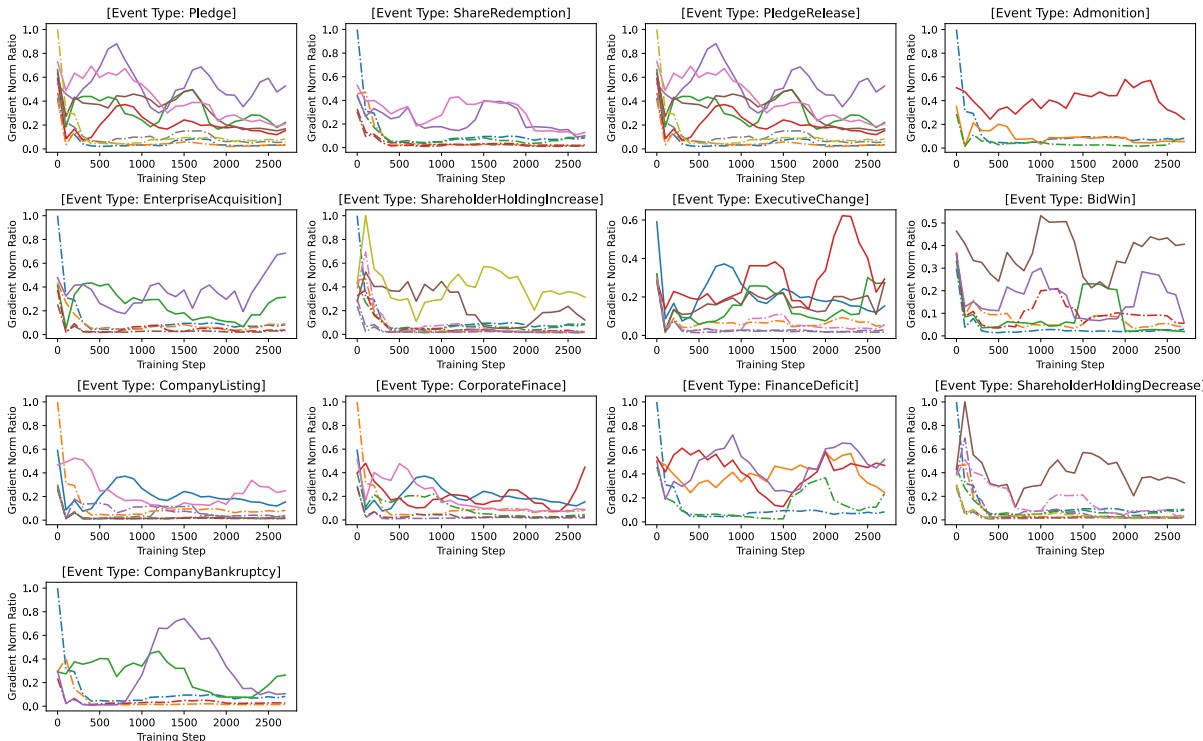

Figure 10: Gradient Norm Ratio Analysis of IPGPF-*without pre-filling* Model on DuEE-fin Training Set. The plot depicts the gradient norm ratio of the queries associated with each event role in the DuEE-fin dataset during the initial 3,000 training steps. The solid line represents event roles with non-zero accuracy on the dev/test set, while the dashed line denotes event roles with zero accuracy.

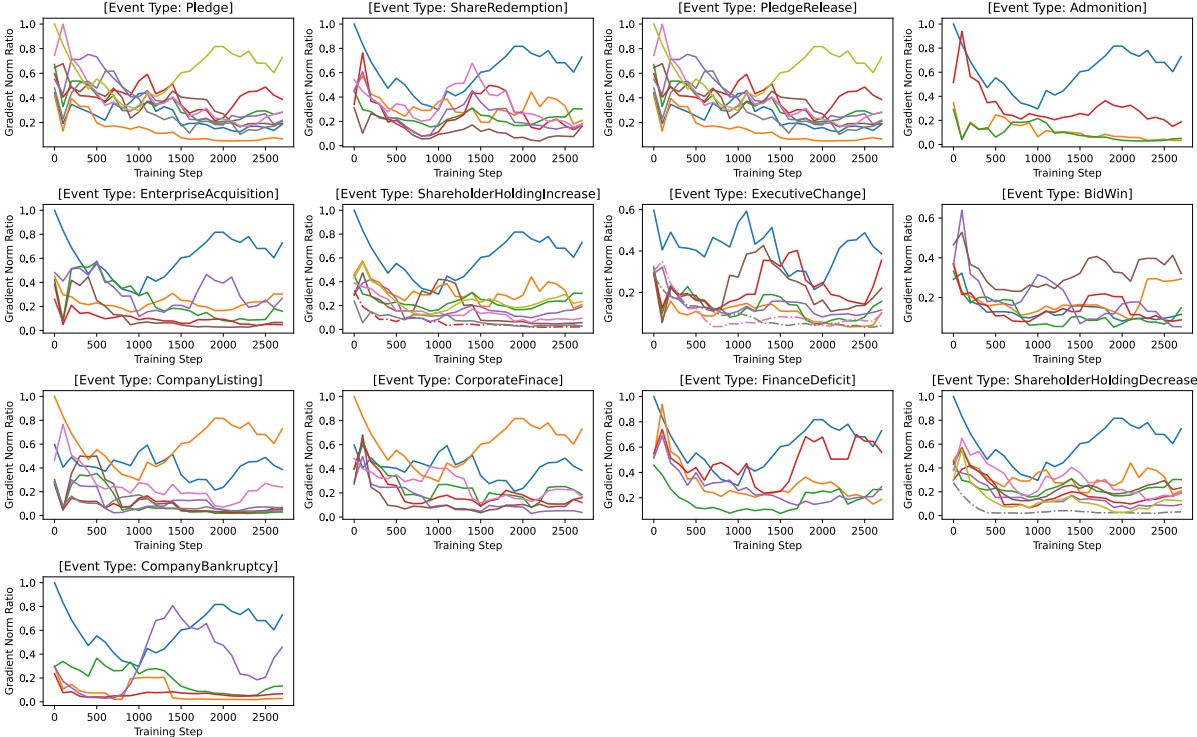

Figure 11: Gradient Norm Ratio Analysis of our IPGPF Model on DuEE-fin Training Set. The plot depicts the gradient norm ratio of the queries associated with each event role in the DuEE-fin dataset during the initial 3,000 training steps. The solid line represents event roles with non-zero accuracy on the dev/test set, while the dashed line denotes event roles with zero accuracy.