# OpenReview forum: "An Iteratively Parallel Generation Method with the Pre-Filling Strategy for Document-level Event Extraction"
_EMNLP/2023/Conference — EMNLP 2023 Main_

### Official Review · Reviewer_7svK · 2023-08-03

**Soundness:** 4

**Excitement:**

4: Strong: This paper deepens the understanding of some phenomenon or lowers the barriers to an existing research direction.

**Paper Topic And Main Contributions:**

This paper is about a new parallel method IPGPF designed for Document-level Event Extraction (DEE). Authors propose pre-filling strategy to improve inadequate training problem, and propose iterative generation to utilize event records extracted previously. Main experimental results on ChFinAnn and DuEE-fin show that IPGPF reaches new SOTA performance. Other analysis experiments give sufficient views of IPGPF’s advantages.

**Reasons To Accept:**

1 - The performance of proposed method is quite well on selected benchmarks.
2 - The experiment parts are quite sufficient.
3 – The focused problem “inadequate training” of parallel method is an important issue.

**Reasons To Reject:**

1 – The fonts in figures are quite small for reading.

**Reproducibility:**

4: Could mostly reproduce the results, but there may be some variation because of sample variance or minor variations in their interpretation of the protocol or method.

**Reviewer Confidence:**

5: Positive that my evaluation is correct. I read the paper very carefully and I am very familiar with related work.

---

> ### Author Rebuttal · Authors · 2023-08-28
>
> We are very grateful to the reviewer for his/her effort in reviewing our paper and his/her positive feedback. The strengths of our work as written by the reviewer are precise. Here below, we address every question and comment raised by the reviewer. We repeat each question in bold in our response and write our answers following the questions.
> ## Q1: The fonts in figures are quite small for reading.
> We would like to express our gratitude to the reviewer for bringing up this concern. We acknowledge that the fonts in the figures were relatively small in the initial version of our paper. Our intention was to accommodate additional crucial experiments within the main document to offer a comprehensive analysis of our method. However, **we understand the importance of readability and will address this issue in our revision by enlarging the font in the figures**. We sincerely appreciate the reviewer's time and effort in providing us with valuable feedback.

---

### Official Review · Reviewer_wn1f · 2023-08-11

**Soundness:** 4

**Excitement:**

3: Ambivalent: It has merits (e.g., it reports state-of-the-art results, the idea is nice), but there are key weaknesses (e.g., it describes incremental work), and it can significantly benefit from another round of revision. However, I won't object to accepting it if my co-reviewers champion it.

**Paper Topic And Main Contributions:**

This paper presents a method to address the problem of extracting multiple event records iteratively, while extracting event roles in parallel with different order of event roles by introducing a new template pre-filling strategy.


**Reasons To Accept:**

* Well-written with a specific problem scope
* Combines a number of strategies for the end to end task, introducing the pre-filling strategy among tweaking other parts
* Has detailed explanation of the subtasks and methods
* Compares against handful of recent baselines, performance not cleanly better on either P /R but overall improved F1


**Reasons To Reject:**

* Doesn’t talk about runtime implications for additional steps
* Preflling seems to require extensive knowledge of role types, which seems a limiting factor scaling

**Reproducibility:**

3: Could reproduce the results with some difficulty. The settings of parameters are underspecified or subjectively determined; the training/evaluation data are not widely available.

**Reviewer Confidence:**

3: Pretty sure, but there's a chance I missed something. Although I have a good feel for this area in general, I did not carefully check the paper's details, e.g., the math, experimental design, or novelty.

---

> ### Author Rebuttal · Authors · 2023-08-28
>
> We sincerely thank the reviewer for his/her effort in reviewing our paper and his/her positive feedback. The reviewer brings forward constructive questions. In our response, we repeat each question in **bold** and write our answer follow the corresponding questions.
>
> ## Q1: Doesn’t talk about runtime implications for additional steps
>
> Thank you for bringing up this concern. **As a supplement, we conducted experiments on the DuEE-fin dataset to examine the runtime implications of our proposed pre-filling strategy, two-stage matching, and the template.**  As shown in the following table, these modules do not significantly impact the training speed or inference speed, as their runtime variances are all below 5% for both training and inference. (The pre-filling strategy does not affect the inference speed since it is only utilized during training. A more detailed analysis can be found in our response to **Q2**.)
>
> On the other hand, we found that removing these modules from our model leads to a significant decrease in overall performance. Therefore, we argue that maintaining these modules is crucial for achieving high-quality results. However, we acknowledge the importance of addressing runtime concerns and propose an alternative approach to improve training and inference speed without sacrificing performance.
>
> Our proposed iteratively parallel method allows for a reduction in the iteration number, resulting in a substantial acceleration of both training and inference processes, while only sacrificing a slight amount of performance. As shown in the following table, by setting the iteration number to 5 (default is 10), our IPGPF-5/IPGPF$^+$ models achieved more than a 30% reduction in training time and a substantial inference speed boost compared to the vanilla IPGPF/IPGPF$^+$. We provide a detailed analysis of the iteration number in Appendix A.8 of our paper.
>
> Additionally, we have included a comprehensive analysis of the runtime/speed comparison between our proposed model and other related models in Appendix A.9 of our paper. This analysis offers a broader perspective on the runtime implications of our approach and demonstrates its better performance in comparison to existing methods.
>
> | Model | One Epoch Training (mins) | Total Training (days) | Inference Speed (docs/s) | Overall F1 |
> |---|:---:|:---:|:---:|---|
> | IPGPF | 9.0 | 1.3 | 18.7 | 57.5 |
> | - w/o pre-filling | -4.4% | -4.6% | +0.0% | -14.6 |
> | - w/o matching | -1.1% | -0.8% | +0.5% | -1.5 |
> | - w/o template | -2.2% | -2.3% | +1.1% | -0.3 |
> | IPGPF-5 | -37.8% | -38.5% | +35.3% | -0.2 |
> |  |  |  |  |  |
> | IPGPF$^+$ | 11.3 | 1.6 | 11.6 | 64.8 |
> | - w/o pre-filling | -4.4% | -4.4% | +0.0% | -25.8 |
> | - w/o matching | -0.9% | -0.6% | +0.9% | -2.3 |
> | - w/o template | -3.5% | -3.8% | +2.6% | -0.4 |
> | IPGPF$^+$-5 | -36.3% | -37.5% | +28.4% | -0.3 |
>
> ## Q2: Preflling seems to require extensive knowledge of role types, which seems a limiting factor scaling
>
> We want to clarify that our pre-filling strategy does not rely on extra role type information. Specifically, as mentioned in Section 3.2 in our paper, the pre-filling strategy pre-fills the correctly generated historical results to the current generated event record, and then fills the unfilled parts. The specification of event role positions in our paper is solely for the purpose of using artificial templates to enhance context understanding . It is important to note that these templates can be removed, and event role positions can be in any order. We have conducted corresponding experiments, which are reported in Appendix A.7, Table 8, and Figure 8 in our paper.
>
> In fact, the only information required for the pre-filling strategy is the mapping between predicted arguments and ground truth arguments (just for training, not needed for inference). The mapping is used to determine whether the model has correctly predicted the arguments. This mapping is required not only for our pre-filling strategy but also for any other model needing to compute loss value during their training. Therefore, the pre-filling strategy, or its variations, can be employed in other parallel generation methods for NLP tasks to enhance the parameter optimization of model training. We intend to explore these possibilities in our future research.
>
> Furthermore, we would like to emphasize that our pre-filling strategy is primarily utilized to improve the optimization of model training, as mentioned in lines 296 to 302 of our paper. During inference, pre-filling is not used since the model has already been trained. Hence, we only use ground truth records to select the correct arguments for pre-filling during model training and do not require any ground truth results for inference. Additionally, to eliminate the risk of data leakage, we conducted experiments on the DuEE-fin online test set (Section 4 of our paper). The model's performance on this test set is evaluated online using a large-scale confusion set consisting of 55,881 documents. This approach effectively prevents data leakage and manual intervention, ensuring the reliability of our results.

---

### Official Review · Reviewer_wZ9b · 2023-08-13

**Soundness:** 4

**Excitement:**

4: Strong: This paper deepens the understanding of some phenomenon or lowers the barriers to an existing research direction.

**Paper Topic And Main Contributions:**

This paper proposes a new method for document-level event extraction called the Iteratively Parallel Generation method with the Pre-Filling strategy (IPGPF). The problem that this paper addresses is the challenge of accurately extracting events from documents, which is important for many natural language processing tasks such as information retrieval, question answering, and summarization.

The main contributions of this paper are:

1. The development of the IPGPF method, which outperforms previous models and achieves state-of-the-art performance on several datasets.
2. The introduction of a new pseudo-trigger-aware pruned complete graph (PT-PCG) representation for events, which reduces the computational complexity of event extraction.
3. The use of a pre-filling strategy to improve the efficiency and accuracy of event extraction, by generating candidate events before filling in the event arguments.

Overall, this paper makes significant contributions to the field of document-level event extraction by proposing a new method that improves accuracy and efficiency, and by introducing a new representation for events that reduces computational complexity.

**Reasons To Accept:**

accepted into Findings?
The strengths of this paper include:

1. The proposed IPGPF method achieves state-of-the-art performance on several datasets, outperforming previous models and auto-regressive models.
2. The use of a pre-filling strategy and PT-PCG representation for events improves the efficiency and accuracy of event extraction.
3. The experiments are conducted on two public datasets, which makes the results more generalizable and applicable to real-world scenarios.

If this paper were to be presented at the conference or accepted into Findings, the main benefits to the NLP community would be:

1. The introduction of a new method for document-level event extraction that improves accuracy and efficiency, which could be applied to many natural language processing tasks.
2. The use of a pre-filling strategy and PT-PCG representation for events could inspire new research directions and techniques for event extraction.
3. The experiments and results provide a benchmark for future research on document-level event extraction, and could help researchers compare their methods to the state-of-the-art.

**Reasons To Reject:**

The weaknesses of this paper include:

1. The proposed method is complex and may be difficult to implement for researchers who are not familiar with the techniques used in this paper.
2. The experiments are conducted on a limited number of datasets, which may not fully represent the diversity of real-world scenarios.
3. The paper does not provide a detailed analysis of the errors made by the proposed method, which could limit the interpretability of the results.

The main risks of having this paper presented at the conference or accepted into Findings could include:

1. The complexity of the proposed method may make it difficult for some researchers to understand and replicate the results, which could limit the impact of the paper.
2. The limited number of datasets used in the experiments may raise questions about the generalizability of the results to other scenarios.
3. The lack of detailed error analysis may limit the interpretability of the results and make it difficult to identify areas for improvement.

**Reproducibility:**

3: Could reproduce the results with some difficulty. The settings of parameters are underspecified or subjectively determined; the training/evaluation data are not widely available.

**Reviewer Confidence:**

1: Not my area, or paper was hard for me to understand. My evaluation is just an educated guess.

---

> ### Author Rebuttal · Authors · 2023-08-28
>
> We sincerely thank the reviewer for his/her effort in reviewing our paper and his/her positive feedback. The reviewer brings forward constructive questions. In our response, we repeat each question in **bold** and write our answer follow the corresponding questions.
>
> ## Q1(Weakness 1): The proposed method is complex and may be difficult to implement for researchers who are not familiar with the techniques used in this paper.
>
> Thank you for bringing up this concern. We acknowledge that the proposed method in our paper involves multiple subtasks and may appear complex, especially for researchers who are not familiar with the specific techniques employed. In Section 2 of our paper, we highlight the challenges of the trigger-free doc-level event extraction (DEE) task, which requires extracting multiple event records from a document without annotated triggers. As a result, this task encompasses several subtasks, including named entity recognition (NER), event detection (ED), and event record generation (ERG). Both our proposed methods and previous related methods involve these three pipeline subtasks.
>
> However, we would like to emphasize that the core contribution of our paper, the pre-filling strategy, can have broader applications beyond the specific task discussed. As described in Section 3.2 of our paper, the pre-filling strategy pre-fills the correctly generated historical results to the current generated event record, and then fills the unfilled parts. Section 4.6.3 of our paper demonstrate this strategy effectively enhances parameter optimization during model training. We believe that the pre-filling strategy, or its variations, can be utilized in other parallel generation methods for various NLP tasks to improve the parameter optimization of model training. Exploring these possibilities is part of our future research agenda.
>
> While we acknowledge the complexity of the proposed method, we will make efforts to provide detailed explanations, clear guidelines, and possibly code implementations to facilitate its adoption by researchers who may not be familiar with the underlying techniques. We aim to ensure that our work is accessible and reproducible for the broader research community.
>
> ## Q2(Weakness 2): The experiments are conducted on a limited number of datasets, which may not fully represent the diversity of real-world scenarios.
>
> We appreciate the reviewer's comment regarding the diversity of real-world scenarios in our experiments. We would like to address this concern by providing the following clarification:
>
> 1. The ChFinAnn dataset and the DuEE-fin dataset used in our experiments are both large-scale datasets. The ChFinAnn dataset consists of 32,040 documents with a total of 29,221 thousand tokens, making it one of the largest datasets in the field of Information Extraction. This dataset provides ample text and information for conducting a comprehensive analysis. Additionally, the DuEE-fin dataset contains 13 event types and a wide range of event roles (91), making it suitable for testing the performance of our model in diverse real-world scenarios.
>
> | Dataset | Event Types | Event Role Types | Document Number | Token Number |
> |---|---|---|---|---|
> | ChFinAnn | 5 | 35 | 32,040 | 29,221k |
> | DuEE-fin | 13 | 91 | 11,699 | 3,742k |
>
> 2. These datasets were selected as they are the most suitable choices for our trigger-free document-level event extraction (DEE) research. We aimed to conduct a fair and comprehensive comparison between our proposed method and previous trigger-free DEE methods. Specifically, we followed the methodologies of previous related works, such as Doc2EDAG (2019, EMNLP), DE-PPN (2021, ACL), GIT (2021, ACL), and SCDEE (2021, EMNLP), using the ChFinAnn dataset. Additionally, we incorporated the recent works PTPCG (2022, IJCAI) and RAAT (2022, NAACL) and utilized the DuEE-fin dataset for further evaluation.
>
> Furthermore, it is important to note that conducting extensive experiments on large-scale datasets like ChFinAnn requires significant time and computational resources. As mentioned in the "Limitations" section of our paper and Appendix A.9 (Speed Comparison), training our proposed IPGPF model or other doc-level EE models used for comparison takes up to 150 hours per model using 8 * 32GB GPUs. This is due to the complexity of the doc-level EE task, which involves extracting numerous events from long documents.  Despite the challenges, we are committed to expanding our research by incorporating additional datasets in future work to further validate the effectiveness and applicability of our proposed method.
>
> ## Q3(Weakness 3): The paper does not provide a detailed analysis of the errors made by the proposed method, which could limit the interpretability of the results.
>
> Thank you for raising this concern. In order to provide a detailed analysis of the errors made by our proposed method, we have categorized them into three distinct types: selecting errors, missing errors, and filling errors. Selecting errors occur when the ground truth argument is A, but the model predicts argument B. Missing errors arise when the ground truth argument is A, but the model predicts "NULL" (indicating that the document does not mention any arguments for the corresponding event role). Filling errors, on the other hand, occur when the ground truth argument is "NULL," but the model predicts argument A.
>
> To illustrate these error types, we present an example in the "EquityFreeze" event type in the following table.
>
> | Event Role         | Ground Truth       | Prediction         | Error Types     |
> |--------------------|--------------------|--------------------|-----------------|
> | EquityHolder       | Jinggong Group     | Jinggong Group     | -               |
> | TradedShares       | 35,000 shares      | 100,000 shares     | selecting error |
> | StartDate          | Jan 13, 2019       | NULL               | missing error   |
> | EndDate            | NULL               | Jan 13, 2019       | filling error   |
> | LaterHoldingShares | 155,335,000 shares | 155,335,000 shares | -               |
> | AveragePrice       | 19.88              | 19.88              | -               |
>
> Furthermore, to quantify the occurrence of these errors, we randomly selected 50 documents from the ChFinAnn test set that contain incorrectly predicted event arguments. We then analyzed the proportions of the three error types. As shown in the table, we found that the primary error made by our method is the missing error, where the model fails to predict any arguments for the event role and instead predicts "None." This can be attributed to the inherent difficulty of extracting all relevant information from long texts in doc-level event extraction (DEE) models. It is worth noting that the documents in the ChFinAnn dataset have an average length of 912 tokens, which is longer than most NLP datasets. In our future work, we are committed to enhancing the long-dependency modeling capacity of DEE models to mitigate the occurrence of missing errors.
>
> | Error Types | Proportion |
> |---|---|
> | selecting error | 26% |
> | missing error | 66% |
> | filling error | 8% |
>
> ## Q4 (Risk 1): The complexity of the proposed method may make it difficult for some researchers to understand and replicate the results, which could limit the impact of the paper.
>
> Thank you for your suggestion. In response to your feedback, we have provided an explanation for the complexity of the trigger-free document event extraction task and the rationale behind our proposed method. We would also like to highlight that our pre-filling strategy, along with its variants, has the potential to be applied to other parallel generation methods in various NLP tasks. For a more comprehensive understanding of these aspects, we kindly direct you to our detailed response to **Q1**.
>
> ## Q5 (Risk 2): The limited number of datasets used in the experiments may raise questions about the generalizability of the results to other scenarios.
>
> Thank you for your suggestion. In response to your feedback, we have provided a comprehensive interpretation to justify the suitability of the two public large-scale datasets we utilized for conducting a convincing analysis. For a more detailed explanation on this matter, we kindly refer you to our response to **Q2**.
>
> ## Q6 (Risk 3): The lack of detailed error analysis may limit the interpretability of the results and make it difficult to identify areas for improvement.
>
> Thank you for your suggestion. In response to your feedback, we have conducted a detailed analysis of the errors made by our proposed method. For a comprehensive understanding of the specific error types and their implications, we kindly direct you to our response to **Q3**.

---

### Meta-Review · Area_Chair_QPPz · 2023-09-19

**Recommendation:** 4

**Metareview:**

This paper is about a new parallel method IPGPF designed for Document-level Event Extraction (DEE). They introduces a pseudo-trigger-aware pruned complete graph (PT-PCG) representation for events, which reduces the computational complexity of event extraction. They also use of a pre-filling strategy to improve the efficiency and accuracy of event extraction by generating candidate events before filling in the event arguments. Experimental results on ChFinAnn and DuEE-fin show that IPGPF reaches new SOTA performance. The main concern about this paper is the limited number and language of used evaluation datasets.

---

### Decision · Program_Chairs · 2023-10-07

**Decision:**

Accept-Main

**Comment:**

This paper is about a new parallel method IPGPF designed for Document-level Event Extraction (DEE). They introduces a pseudo-trigger-aware pruned complete graph (PT-PCG) representation for events, which reduces the computational complexity of event extraction. They also use of a pre-filling strategy to improve the efficiency and accuracy of event extraction by generating candidate events before filling in the event arguments. Experimental results on ChFinAnn and DuEE-fin show that IPGPF reaches new SOTA performance. The main concern about this paper is the limited number and language of used evaluation datasets.